# Protective effects of Pt-N-C single-atom nanozymes against myocardial ischemia-reperfusion injury

Tianbao Ye[1,2,7], Cheng Chen[3,7], Di Wang[1,4,7], Chengjie Huang[2], Zhiwen Yan[5], Yu Chen[1], Xian Jin[1]✉, Xiuyuan Wang ✉[6] ✉, Xianting Ding ✉[2] ✉ & Chengxing Shen[1] ✉

Effective therapeutic strategies for myocardial ischemia/reperfusion (I/R) injury remain elusive. Targeting reactive oxygen species (ROS) provides a practical approach to mitigate myocardial damage following reperfusion. In this study, we synthesize an antioxidant nanozyme, equipped with a single-Platinum (Pt)-atom (PtsaN-C), for protecting against I/R injury. PtsaN-C exhibits multiple enzyme-mimicking activities for ROS scavenging with high efficiency and stability. Mechanistic studies demonstrate that the excellent ROS-elimination performance of the single Pt atom center precedes that of the Pt cluster center, owing to its better synergistic effect and metallic electronic property. Systematic in vitro and in vivo studies confirm that PtsaN-C efficiently counteracts ROS, restores cellular homeostasis and prevents apoptotic progression after I/R injury. PtsaN-C also demonstrates good biocompatibility, making it a promising candidate for clinical applications. Our study expands the scope of single-atom nanozyme in combating ROS-induced damage and offers a promising therapeutic avenue for the treatment of I/R injury.

Acute myocardial infarction remains the leading cause of death globally, with an increasing trend among younger individuals in recent years[1]. Sudden occlusion of the coronary artery results in severe death of cardiomyocytes. Timely revascularization therapy is crucial for minimizing myocardial damage and improving long-term clinical outcomes[2]. Paradoxically, revascularization itself can lead to cardiomyocyte death and exacerbate myocardial injury after ischemia, which is known as myocardial ischemia/reperfusion (I/R) injury[3,4]. The mechanisms of I/R injury are complex and involve various factors, including oxidative stress (OS), calcium overload, energy metabolism dysfunction, and activation of inflammatory cells[5]. However, current therapeutic strategies to mitigate I/R injury have shown limited success[2,6,7]. Therefore, there is an urgent need for effective therapeutic approaches to alleviate I/R injury.

Reactive oxygen species (ROS) is considered as a most predominant factor for I/R injury[5,6]. The ROS burst is mainly attributed to the electron leakage mediated by impaired electron transport chain and reduced oxidative phosphorylation activity in mitochondria during I/R injury[8,9]. Excessive ROS induces oxidative damage to DNA, proteins, and liquids, ultimately resulting in cell death[10,11]. Although endogenous antioxidant enzymes including superoxide dismutase (SOD), glutathione peroxidase (GPx), catalase (CAT), and peroxidase (POD) work to maintain the homeostasis, the burst of ROS can overwhelm and wreck the antioxidant

[1]Department of Cardiology, Shanghai Sixth People's Hospital Affiliated to Shanghai Jiao Tong University School of Medicine, 200233 Shanghai, China. [2]Institute for Personalized Medicine, School of Biomedical Engineering, Shanghai Jiao Tong University, 200030 Shanghai, China. [3]Tongji Hospital, School of Medicine, Tongji University, 200092 Shanghai, China. [4]Department of Cardiology, Shanghai General Hospital, Shanghai Jiao Tong University School of Medicine, 200080 Shanghai, China. [5]Youth Science and Technology Innovation Studio of Shanghai Jiao Tong University School of Medicine, 200233 Shanghai, China. [6]Zhongshan Hospital, Fudan University, 200032 Shanghai, China. [7]These authors contributed equally: Tianbao Ye, Cheng Chen, Di Wang. ✉e-mail: jinxiannian@126.com; wangxiuyuan95@yeah.net; dingxianting@sjtu.edu.cn; shencx@sjtu.edu.cn

systems[9–13]. Thus, targeting ROS scavenging represents a promising therapeutic approach.

Currently, extensive research has focused on strategies, including the inhibitors of ROS sources and antioxidants, to improve redox balance[14–16]. Recently, many clinical trials have demonstrated the highly beneficial effects of guanylyl cyclase enzyme activators, xanthine oxidase inhibition, and isosorbide dinitrate/hydralazine, however, there remain no effective antioxidants approved for clinical treatment of I/R injury[17–19]. Despite a few widely used cardiovascular drugs with potential antioxidant effects, including angiotensin-converting enzyme inhibitors, β-blockers, and coenzyme Q10, the slow onset of action and limited treatment effectiveness hinder their widespread application as antioxidats[18,20]. This hindrance is attributed to the poor druggability, low drug bioavailability, and non-negligible side effects[19,21].

To overcome these challenges, artificial nanozymes have emerged as a promising alternative[22]. Nanozymes, possessing the advantage of good druggability, efficient catalytic activity, and high biosafety, have exhibited remarkable therapeutic effects against I/R injury[23,24]. Among the various nanozymes, single-atom nanozymes (SANs) have garnered significant attention due to their high catalytic activity, maximized metal utilization, and remarkable selectivity[25,26]. The well-defined coordination structure of metal-nitrogen-carbon, with similar active sites of natural metalloenzymes, greatly amplifies the catalytic property with high sensitivity[26,27]. Platinum (Pt), with the advantage of high cytocompatibility, stability, and multiple enzyme-like activities, has been considered as a substitute for natural enzymes to combat oxidative stress[28–30]. As the catalytic performance of Pt highly depends on the particle diameter, decreasing the size of Pt particle significantly boosts catalytic activity[31–33]. Therefore, it is promising to construct a SAN doped with Pt to maximize the efficacy of ROS scavenging while minimizing the risk of metal cytotoxicity, making it an ideal approach for myocardial I/R therapy.

In this work, we design a Pt single-atom nanozyme with a structure of Pt-$N_4$-C (PtsaN-C) to enhance its antioxidative functions in eliminating ROS during myocardial I/R injury. The PtsaN-C nanozyme not only inherits the advantage of Pt nanoparticles (PtNPs) with respect to multienzyme-mimicking properties, high stability, and good biocompatibility but also possesses high catalytic performance and selectivity. Our results demonstrate that PtsaN-C exhibits more efficient catalytic activity and utilization of Pt atoms compared with the structure of Pt nanoparticles-based $N_4$-C (PtnpN-C) (Fig. 1). Density functional theory (DFT) calculations suggest that the remarkable catalytic performance of PtsaN-C can be attributed to both well synergistic effect and metallic electronic structure. In vitro experiments show that PtsaN-C possesses excellent cytoprotection by scavenging ROS and reducing cellular apoptosis induced by oxygen–glucose deprivation/reoxygenation (OGD/R) injury. Subsequent in vivo experiments reveal that PtsaN-C significantly diminishes cardiac injury, reduces infarct volume, and improves cardiac function post-myocardial I/R injury. Besides, systematic evaluations confirm the good biocompatibility of PtsaN-C. Mechanistically, proteomics results indicate that PtsaN-C exerts multiple biological functions, particularly inhibiting the mitogen-activated protein kinase/ c-Jun N-terminal kinase (MAPK/Jnk) signaling pathway to provide cardioprotection. Overall, this study introduces a nanoreactor with multiple antioxidant enzyme-like properties and high efficiency in clearing ROS, which also minimizes the biotoxicity associated with traditional Pt nanozymes. These findings offer a promising therapeutic alternative for the treatment of myocardial I/R injury.

## Results

### Design and structural characterization of PtsaN-C

We synthesized atomic dispersed Pt biocatalysts, referred to as PtsaN-C, by using graphene quantum dots (GQDs) and Pt salt solutions for precursors, as previously reported[34] (Fig. 2a and Supplementary Table 1). Additionally, PtnpN-C, consisting of Pt metal nanoparticles, and N-doped GQDs without metal sites (N-C), were synthesized as reference biocatalysts. Through high-resolution transmission electron microscopy (HR-TEM) and atomic force microscopy (AFM), we confirmed that the average particle size of GQDs was about 5–8 nm (Fig. 2b). The lattice fringes of 0.240 nm were identified as the typical spacing of GQDs. Aberration-corrected high-angle annular dark field scanning TEM (HAADF-STEM) was used to identify the Pt distribution (Fig. 2c). Isolated Pt atoms appeared as bright dots due to their higher Z contrast compared to the C/N sites. Energy dispersive spectrometer (EDS) mapping further confirmed the high Pt dispersion on the biocatalyst (Fig. 2d and Supplementary Fig. 1). Moreover, HR-TEM images of PtnpN-C revealed uniformly loaded Pt nanoparticles with diameters of 5–10 nm on the GQDs support (Fig. 2e). The Pt (100) plane could be identified by assigning the enlarged lattice fringes (0.209 nm) (Fig. 2f). Additionally, analysis of N2 physical adsorption–desorption isotherms demonstrated a typical type IV isotherm with an H3-type hysteresis loop, indicating that the addition of Pt did not affect the physical adsorption properties and pore structure characteristics of GQDs (Fig. 2g).

### Structural analysis of atom and composition of PtsaN-C

To determine the structural characteristics of Pt in PtsaN-C and PtnpN-C, we performed high-resolution X-ray Diffraction (XRD) analysis in the range of 37–47° (Fig. 3a). The observed 39.7° and 46.54° on PtnpN-C were assigned to Pt (111) and Pt (100), respectively (Supplementary Fig. 2, Supplementary Table 2). Moreover, the broad peak observed at around 43° belonged to the diffraction signal of amorphous carbon. In contrast, no diffraction signal of Pt was detected in the PtsaN-C sample, confirming the atomic-scale dispersion of Pt on the PtsaN-C surface. To further confirm the dispersion of Pt, we utilized attenuated total refraction-Fourier-transform infrared spectroscopy (ATR-FTIR) with CO molecule as a probe (Fig. 3b). When Pt clusters were presented on the catalyst surface, CO adsorbed on Pt in two configurations including bridge adsorption and linear adsorption, which could be detected at approximately 1860 and 2070 cm$^{-1}$, respectively. The absence of a vibration band at around 1860 cm$^{-1}$ indicated that Pt was bound to the substrate monatomically. To gain insights into the valence information of the two Pt-based samples, Pt *4f* and N *1s* deconvolution of X-ray photoelectron spectroscopy (XPS) was performed (Supplementary Fig. 3, Supplementary Fig. 4). The high binding energies of 72 eV and 76 eV were generally assigned to Pt$^{2+}$, while the lower binding energy peaks, indicating the presence of Pt$^0$, were solely observed in PtnpN-C (Fig. 3c, Supplementary Table 3). In N *1s*, N species at high binding energy were doped on the support in the form of graphitic N, generally considered to be catalytically inert. Previous studies have identified pyridine N at 398.5 eV as an excellent substrate for anchoring metal single atoms[35]. Consistently, we observed a higher proportion of pyridinic N in PtsaN-C (Fig. 3d and

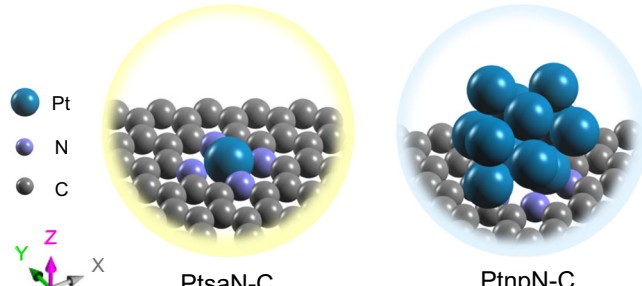

**Fig. 1 |** Schematic diagram of Pt single atom (PtsaN-C) and Pt nanoparticles (PtnpN-C) on graphene quantum dots.

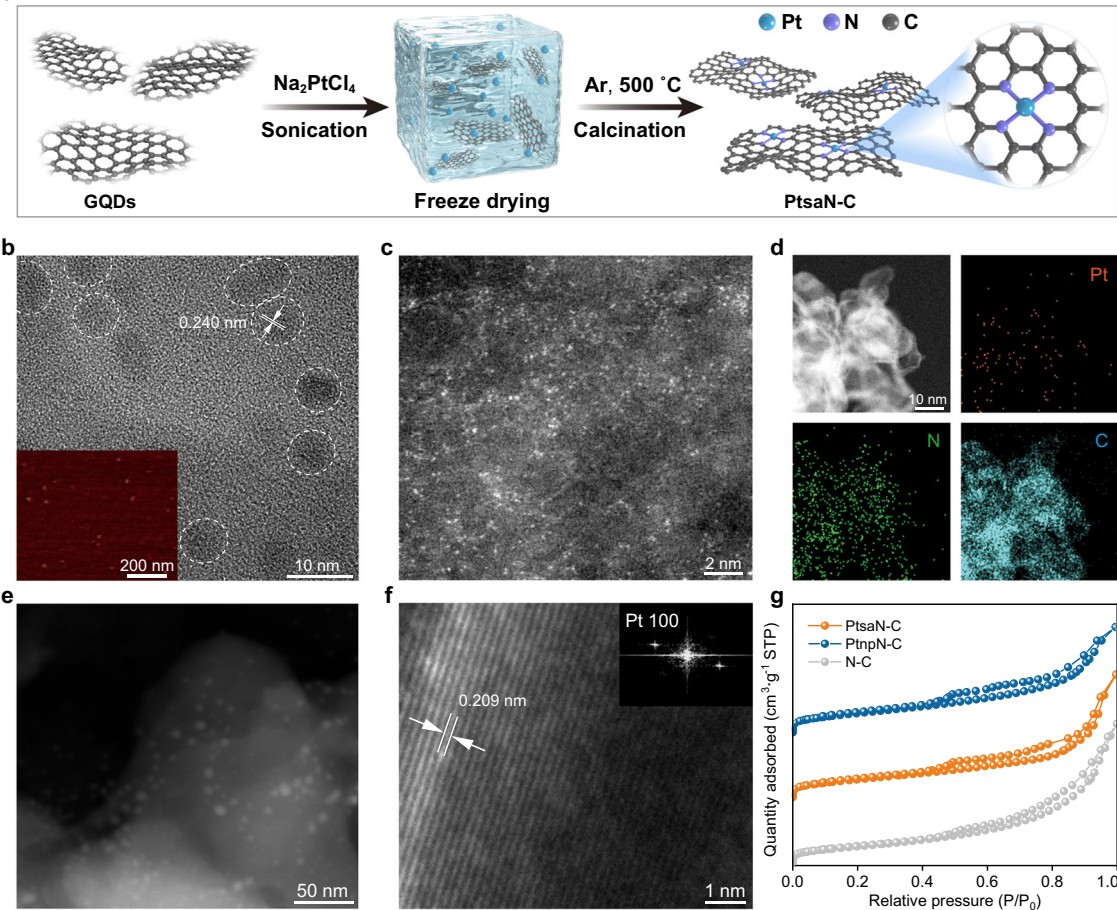

**Fig. 2 | Preparation and characterization of PtsaN-C. a** Schematic illustration of the preparation strategy for PtsaN-C. **b** Typical HR-TEM image of GQDs with a uniform lateral size of approximately 5–8 nm (scale bar: 10 nm). **c** HAADF-STEM image of PtsaN-C, where the bright dots correspond to Pt single atoms (scale bar: 2 nm). **d** EDS mapping images of PtsaN-C (scale bar: 10 nm). **e** HR-TEM image of PtnpN-C (scale bar: 50 nm). **f** HR-TEM image and fast Fourier transform pattern of PtnpN-C in the inset (scale bar: 1 nm). **g** $N_2$ adsorption–desorption isotherms of PtsaN-C, PtnpN-C, and N-C. Experiments were conducted twice (**b**) or three times (**c**, **e**, **f**) with similar results. Source data are provided as a Source Data file.

Supplementary Table 4). We further investigated the valence state and coordination of Pt in PtsaN-C using X-ray absorption spectroscopy (XAS). The white line peak of PtsaN-C was found between those of Pt and $PtO_2$, suggesting partial charge transfer from the Pt single atom to the adjacent metalloid coordination (Fig. 3e). Additional analysis using the Fourier-transformed extended X-ray absorption fine structure (FT-EXAFS) spectra in R-space confirmed Pt-N coordination in PtsaN-C (Fig. 3f). The fitted FT-EXAFS structural parameters indicated an average coordination number of approximately 4 for Pt–N shell in PtsaN-C (Fig. 3g, Supplementary Fig. 5, Supplementary Table 5). Therefore, the atomic structure of the Pt species in the PtsaN-C biocatalyst was proposed to be the Pt-$N_4$ configuration, as shown in the inset of Fig. 3g. The two-dimensional spectrum after wavelet transformation was also conducted to assess the coordination environment of Pt atoms. The strong signals at $R = \sim 1.5$ Å and $k = \sim 4.5$ Å came from the coordination of Pt and light-mass N atoms (Fig. 3h). Collectively, our precise spectroscopic analysis confirms the existence of abundant and uniform Pt-$N_4$ sites in PtsaN-C.

### Multienzyme-mimicking properties of PtsaN-C

The ROS-scavenging and multienzyme-mimicking properties of PtsaN-C were systematically evaluated. SOD, a crucial antioxidant enzyme of cellular defense against ROS, functions by converting $O_2^{\cdot-}$ into $H_2O_2$ and $O_2$[36]. This enzymatic activity can be measured using WST-8, which reacts with $O_2^{\cdot-}$ to produce formazan that can be

quantified at 450 nm. Comparative analysis with the references revealed that PtsaN-C exhibited an excellent $O_2^{\cdot-}$ metabolism rate even at a low dosage, suggesting remarkable SOD-like activity (Fig. 4a). Similarly, the absorbance changes of formazan over time further confirmed the strong SOD-like activity of PtsaN-C (Supplementary Fig. 6).

CAT-like activity is well known for the ability to decompose $H_2O_2$ into $O_2$ and $H_2O$. As $\cdot$OH from $H_2O_2$ can react with terephthalic acid (TPA) to form fluorescent 2-hydroxy terephthalic acid, the CAT-like activity can be evaluated by fluorescence intensity of 2-hydroxy terephthalic acid. When PtsaN-C was present, the content of $H_2O_2$ experienced a sharp decrease, as evidenced by significantly lower fluorescent intensity compared to the references, highlighting the excellent CAT-like property of PtsaN-C (Supplementary Fig. 7). The CAT-like activity was further assessed by monitoring the dissolved $O_2$ concentration. Notably, the presence of PtsaN-C led to a rapid increase in $O_2$ concentration compared to the other groups (Fig. 4b). Moreover, adding PtsaN-C to the $H_2O_2$ solution resulted in the generation of more bubbles, indicating the potent ability of PtsaN-C to decompose $H_2O_2$ into $H_2O$ and $O_2$ (Supplementary Fig. 8a). Meanwhile, although PtnpN-C also exhibited CAT-like activity, its catalytic performance was less effective compared with PtsaN-C. Furthermore, steady-state kinetics analysis revealed that PtsaN-C exhibited a higher maximum reaction velocity ($V_{max}$=5.343 mg·$L^{-1}$·$min^{-1}$) and a lower catalytic constant ($K_m$ = 19.33 mM) as compared to PtnpN-C ($V_{max}$=3.103 mg·$L^{-1}$·$min^{-1}$,

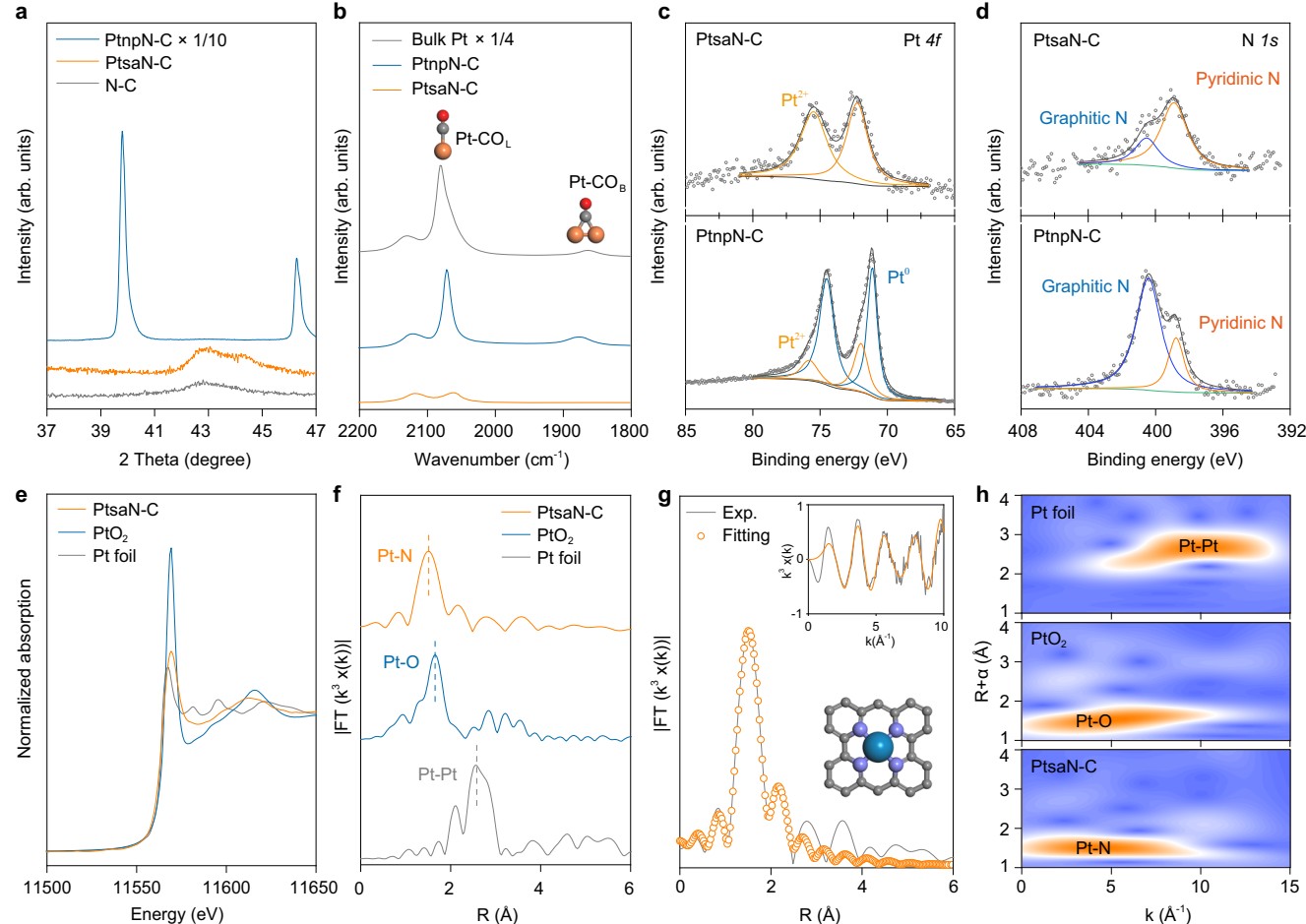

**Fig. 3 | Structural analysis of atom and composition of PtsaN-C. a, b** High-resolution XRD patterns (**a**) and FTIR spectra of CO adsorption (**b**) for the PtsaN-C, PtnpN-C, and N-C samples. **c, d** Core-level XPS spectra for Pt $4f$ (**c**) and N $1s$ region (**d**) of PtsaN-C and PtnpN-C. **e** Normalized X-ray absorption near edge structure (XANES) spectra at the Pt L3-edge for Pt foil, PtO$_2$, and PtsaN-C. **f** The k2-weighed FT-EXAFS spectra for Pt foil, PtO$_2$, and PtsaN-C in the R-space. **g** The R-space fitting results of PtsaN-C, with the k-space fitting results and proposed Pt-N$_4$ coordination structure included as insets. **h** Wavelet transformation images of the EXAFS signals for Pt foil, PtO$_2$, and PtsaN-C. Source data are provided as a Source Data file.

$K_m = 32.09$ mM) (Supplementary Fig. 8b–d). Collectively, these findings confirmed the excellent CAT-like activity of PtsaN-C.

Iron-based nanomaterials have been reported to exhibit significant POD-like activity by decomposing H$_2$O$_2$ into H$_2$O. The assessment of POD-like activity can be achieved using 3,3′,5,5′-Tetramethylbenzidine (TMB), which is converted into blue oxidized TMB with a characteristic absorbance at 652 nm in the presence of H$_2$O$_2$ and peroxidase. The PtsaN-C group displayed a darker blue color compared to the PtnpN-C group, indicating that PtsaN-C possessed a higher POD-like activity (Supplementary Fig. 9a). Furthermore, the absorption spectrum revealed higher absorbance in the PtsaN-C disposal (Fig. 4c and Supplementary Fig. 9b). Additionally, the POD-like activity of PtsaN-C towards H$_2$O$_2$ substrates followed typical Michaelis−Menten kinetics (Supplementary Fig. 10a). PtsaN-C displayed a larger $V_{max}$ (1.231 mM·s$^{-1}$) compared to PtnpN-C (0.675 mM·s$^{-1}$), while the $K_m$ of PtsaN-C (2.943 mM) was lower than that of PtnpN-C (3.658 mM) (Supplementary Fig. 10b, c). These results indicated the distinguishing POD-like activity of PtsaN-C.

GPx, an endogenous enzyme responsible for defending against ROS, can effectively decompose H$_2$O$_2$ into H$_2$O with the assistance of GSH. Our results revealed that the GPx-like property of PtsaN-C was superior to PtnpN-C (Fig. 4d). Michaelis−Menten kinetic analysis further demonstrated that the $V_{max}$ of PtsaN-C (15.52 μM·min$^{-1}$) was more than 1.5 times higher than that of PtnpN-C (9.32 μM·min$^{-1}$), and the $K_m$ of PtsaN-C (0.035 mM) was lower than PtnpN-C (0.086 mM)

(Supplementary Fig. 11). Additionally, electron paramagnetic resonance (EPR) spectra demonstrated the effective ·OH scavenging capacity of PtsaN-C (Fig. 4e). Taken together, these results provided robust evidence that PtsaN-C possessed multienzyme-mimicking antioxidant properties.

Natural enzymes present certain inherent limitations, such as difficulties in recycling, storage, thermal, and *pH* labile. To assess stability, PtsaN-C was subjected to comparison with the natural catalase. The catalytic activity of PtsaN-C remained unaffected even with the repetitive addition of H$_2$O$_2$, indicating sustained catalytic ability (Fig. 4f). Comparison to natural catalase, PtsaN-C not only exhibited optimal function over a wider *pH* range (Fig. 4g), but also presented thermostability (Fig. 4h). When compared to recently reported metal-based biocatalysts with respect to CAT-like activity, PtsaN-C displayed a highest affinity and catalytic efficiency, as evident from its $K_m$ and turnover number (TON) values (Fig. 4i and Supplementary Table 6). All these results confirmed the high multienzyme-mimicking antioxidant activity and stability of PtsaN-C.

**In situ Raman and density functional theory calculations of CAT-like activities of PtsaN-C**

To advance the comprehension of the efficient degradation of ROS by PtsaN-C, we employed in situ enhanced Raman spectroscopy. Phorbol myristate acetate (PMA) was utilized to facilitate ROS generation in H9C2 cell lines. Briefly, PtsaN-C and PtnpN-C were added into the

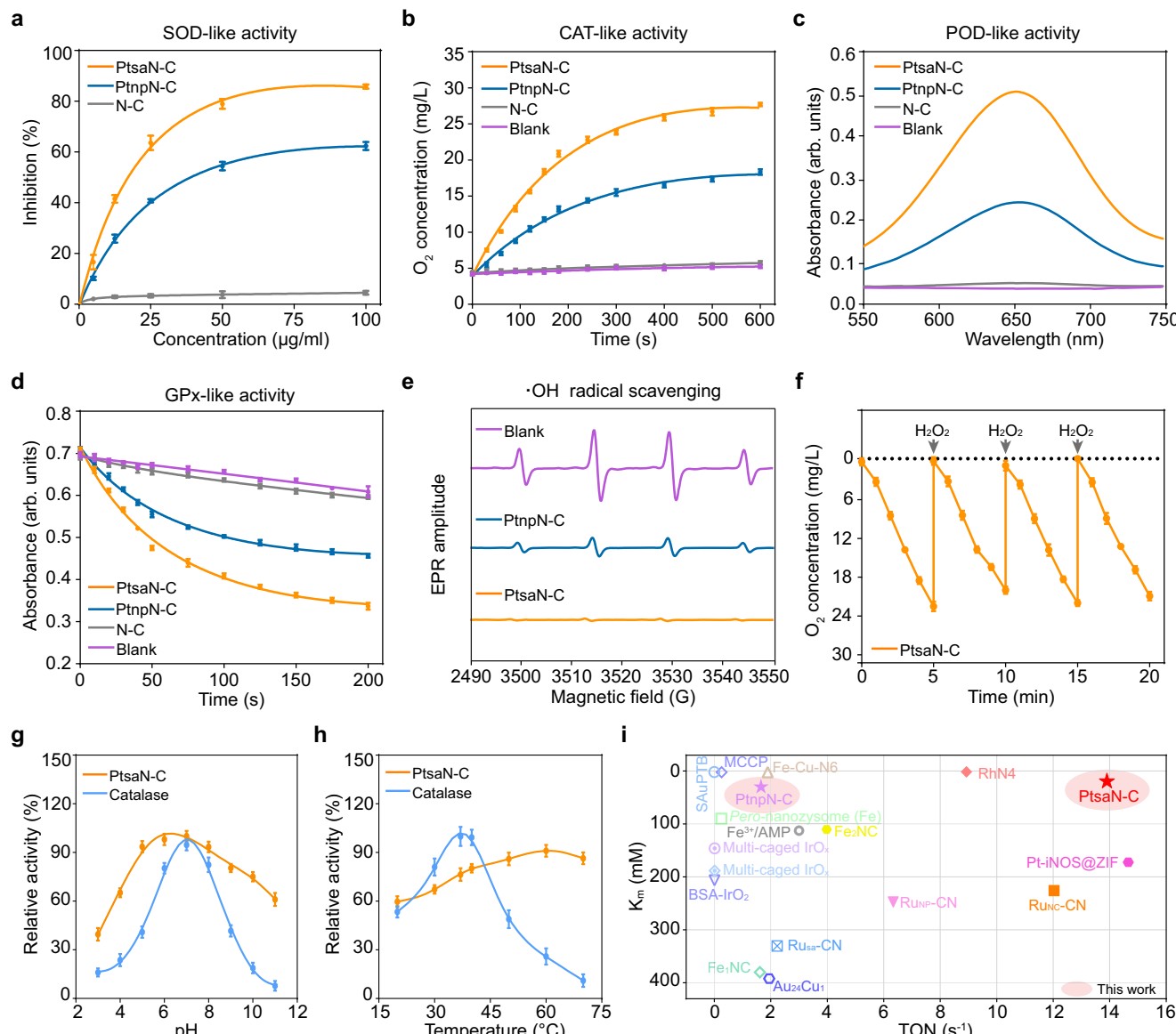

**Fig. 4 | ROS-scavenging and multienzyme-mimicking antioxidant properties of PtsaN-C. a** Concentration-dependent SOD-like activity of PtsaN-C, N-C, and PtnpN-C ($n = 3$ independent samples). **b** Time-dependent CAT-like activity of PtsaN-C and reference biocatalysts ($n = 3$ independent samples). **c** POD-like activity presented with absorbance curves for the related groups. Experiments were repeated three times with similar results. **d** GPx-like activity tests of PtsaN-C, PtnpN-C, N-C, and blank samples ($n = 3$ independent samples). **e** EPR spectra for ·OH scavenging property of PtsaN-C and PtnpN-C. Experiments were repeated three times with similar results. **f** Stability test of the catalytic ability of PtsaN-C with repetitive addition of $H_2O_2$ at different cycles ($n = 3$ independent samples). **g, h** Catalytic activity of PtsaN-C at various pH (**g**) and temperature (**h**) conditions ($n = 3$ independent samples). **i** Comparison of TON and $K_m$ with recently reported nanozymes in terms of CAT-like catalysis. Data are represented with mean ± SEM. Source data are provided as a Source Data file.

culture medium respectively before PMA stimulation. The result showed that Raman signal peaks attributed to *O (-541 cm$^{-1}$), *OOH (-674 and 735 cm$^{-1}$), and *OH (-975 cm$^{-1}$) appeared in the spectrum after PMA stimulation (Fig. 5a). In specimens with PtsaN-C, the signal peaks underwent swift degradation within a 10-min timeframe. Conversely, in samples hosting PtnpN-C, the degradation rate was slower compared with PtsaN-C, and the persistence of *O and *OOH in the milieu was still discernible even after 30 min. This result verified that PtsaN-C possessed a stronger property of free radical scavenging than PtnpN-C.

Density functional theory (DFT) calculations were also performed to elucidate the original activity of Pt-N$_4$ sites in PtsaN-C regarding ROS-scavenging. Three models (PtsaN-C, PtnpN-C, and N-C) were structured and compared to explore the energy and electronic modulation mechanisms (Supplementary Fig. 12). Following a widely

recognized reaction pathway for ROS scavenging based on a previous study[37], the distribution of energy barriers was examined (Fig. 5b and Supplementary Fig. 13). Our findings revealed that PtsaN-C had a smaller reaction energy barrier (0.57 eV), which was significantly lower than that of PtnpN-C (0.90 eV) and N-C (1.61 eV) derived from reaction energy calculations (Fig. 5c, Supplementary Table 7). This result aligned with experimental observations and reaffirmed that the Pt-N$_4$ configuration served as a highly active site for ROS dissociation. Furthermore, an analysis of Bader charge and work function demonstrated that the single Pt atom in Pt-N$_4$ experienced significant electron loss (-0.74 e$^-$) in contrast to Pt nanoparticles (−0.05 e$^-$), which could be attributed to the strong electron attraction from nitrogen in Pt-N$_4$ (Fig. 5d and Supplementary Fig. 14). This resulted in the positive valence state of the single Pt atom on PtsaN-C, enhancing its ability to absorb negative electron groups ($O_2^{·-}$, ·OH, etc.), consistent with in situ

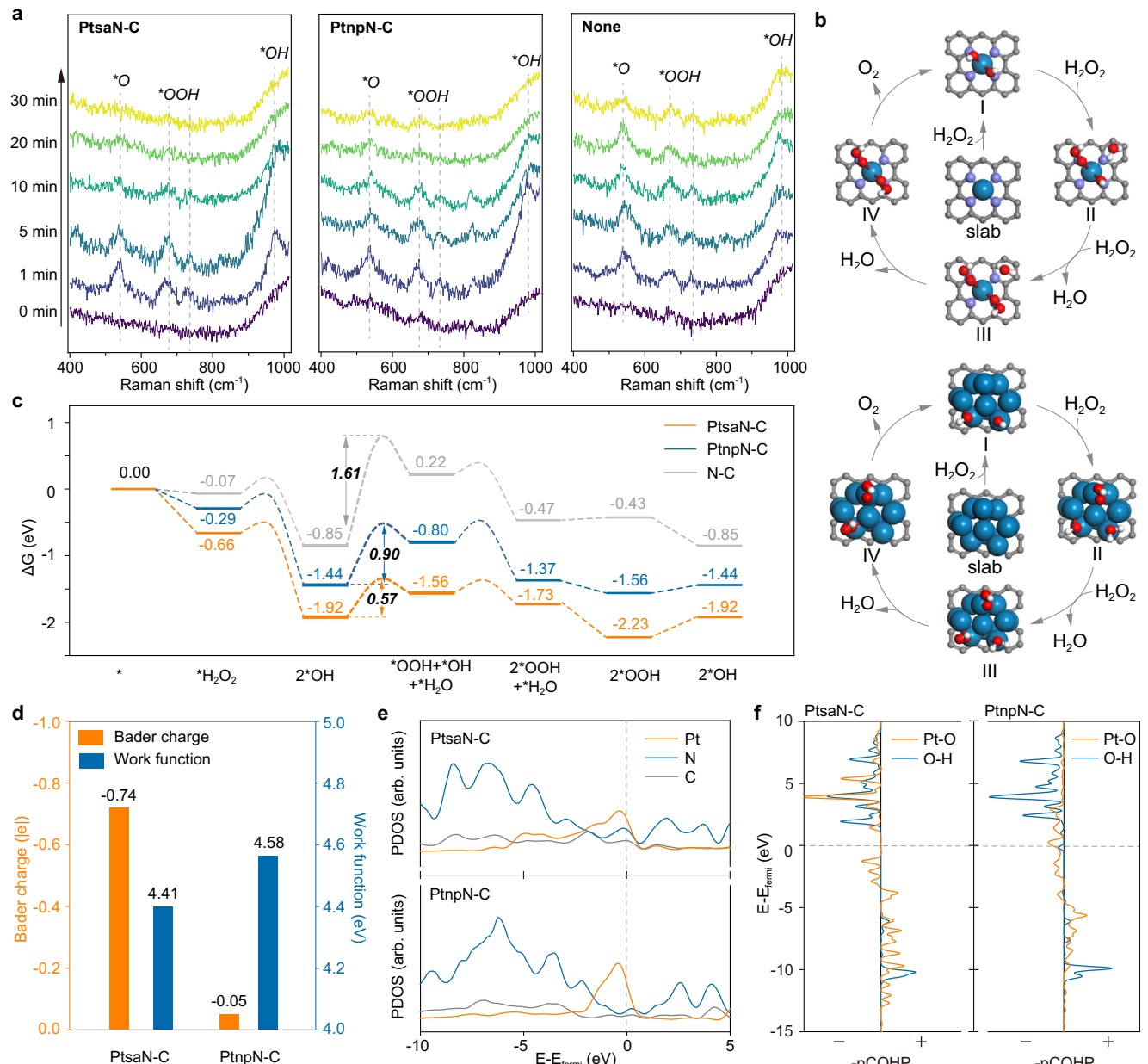

**Fig. 5 | In situ Raman and DFT calculation study on the CAT-like ROS-elimination activities of PtsaN-C. a** In situ SERS spectroscopy observed H9C2 cell lines containing PtsaN-C or PtnpN-C within response time from 0 to 30 min after PMA stimulation. None refers to the H9C2 cell lines without nanoenzymes. **b** Proposed reaction pathways on PtsaN-C and PtnpN-C models. **c** Reaction energy barrier diagram of potential pathways on the PtsaN-C, PtnpN-C, and N-C models. **d** Bader charge and work function of PtsaN-C and PtnpN-C models. **e, f** PDOS (**e**) and COHP (**f**) analysis of PtsaN-C and PtnpN-C models. Source data are provided as a Source Data file.

Raman and X-ray characterization analyses (Figs. 5a and 3c–g). Additionally, partial density of states (PDOS) analysis revealed a decrease in charge density of Pt single-atom active sites near the Fermi level, accompanied by an increase in electron density of C and N (Fig. 5e). This indicated that the activity of Pt was attributed to the Pt-N coordination structure (Supplementary Fig. 15). However, more prominent Pt-Pt coordination in PtnpN-C had a minor contribution to ROS scavenging. Finally, a bonding analysis was performed using the crystal orbital Hamilton population (COHP). It is widely accepted that the formation of Pt-O bonds and the cleavage of O-H occur in the process of $H_2O_2$ adsorption and decomposition on Pt sites. As a result, PtsaN-C exhibited a stronger Pt-O electron distribution and a weaker O-H electron distribution compared to PtnpN-C, suggesting the facilitation of stable Pt-O bond formation in electron-deficient Pt sites (Fig. 5f).

This weakened the bond energy of O-H and accelerated the dissociation of $H_2O_2$.

## The antioxidant and anti-apoptosis ability of PtsaN-C post-OGD/R injury in vitro

Investigations mentioned above have firmly proved the ROS-scavenging property of PtsaN-C with multienzyme-mimicking activity. Our studies revealed that PtsaN-C exhibits good cytocompatibility in a rat cardiomyocyte cell line (H9C2) across a wide range of concentrations, possessing superiority in comparison with bare Pt nanoparticles (Supplementary Fig. 16). We then proceeded to evaluate the potential application of PtsaN-C in protecting cells from OGD/R injury. OGD/R protocol was employed as an in vitro model to mimic I/R injury (Fig. 6a). Remarkably, even at a low concentration of 1 μg/ml, PtsaN-C

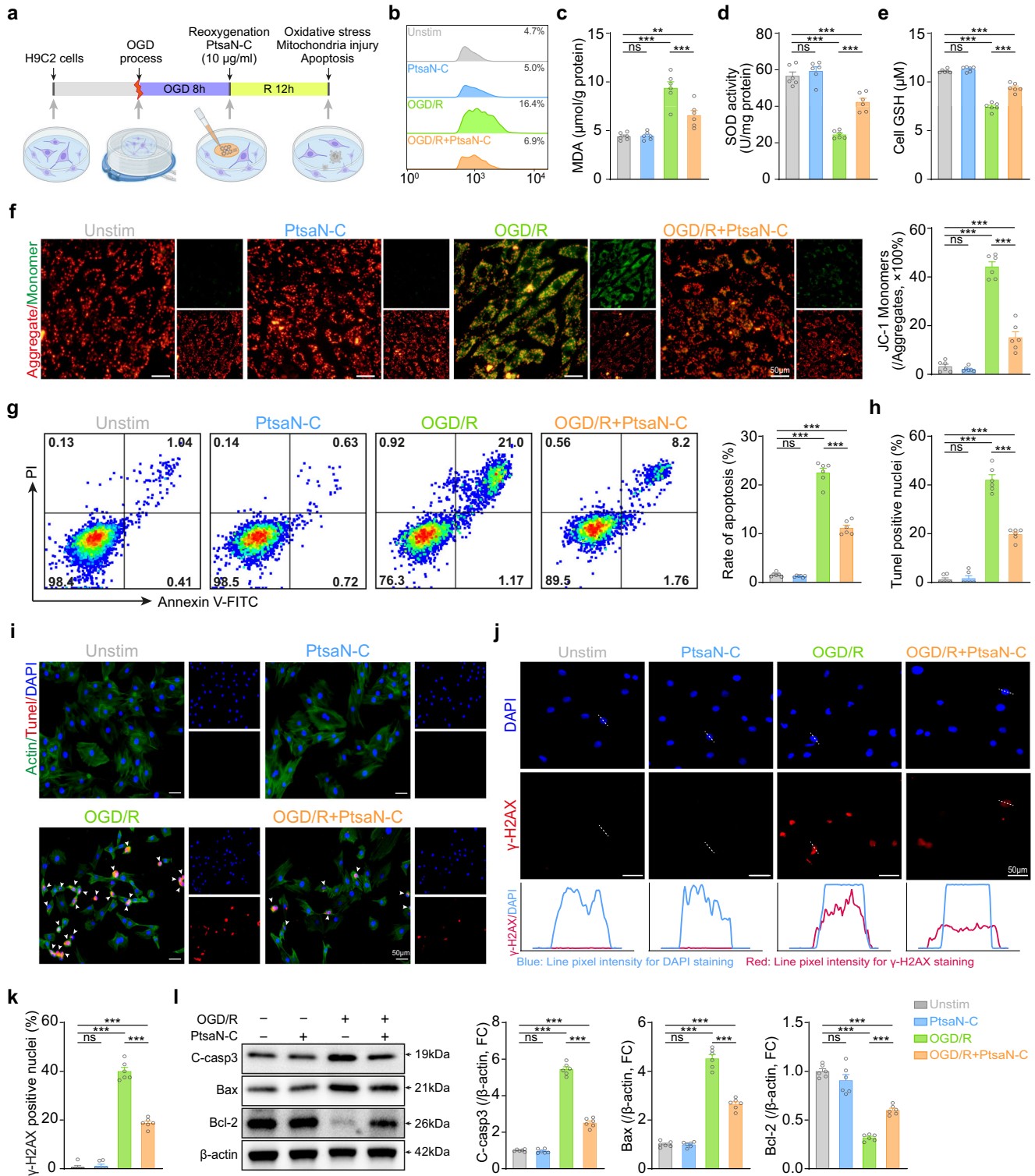

significantly rescued cell viability and protective effects improved with the increase of concentration (Supplementary Fig. 17). However, further increasing the PtsaN-C concentration up to 50 μg/ml did not significantly increase the protective potency compared with 10 μg/ml, suggesting that 10 μg/ml of PtsaN-C effectively quenched most intracellular ROS. Administration of PtsaN-C not only lowered cellular ROS and MDA levels, but restored SOD activity, GSH levels, and ATP content, thereby further confirming the antioxidant capacity of PtsaN-C during the OGD/R process (Fig. 6b–e and Supplementary Fig. 18). In particular, these effects could be more distinctly observed at the few

hours after reoxygenation (Supplementary Fig. 19). Given that mitochondria are the main source of ROS and in turn sensitive to ROS, its damage can typically manifest with reduced mitochondrial membrane potential[38]. Our assessment of mitochondrial transmembrane potential indicated that PtsaN-C could scavenge ROS to restore mitochondrial membrane potential and maintain mitochondrial function, consequently preventing ROS bursts (Fig. 6f). Besides, the impaired cardiomyocyte contractile activity induced by OGD/R injury was restored after treating with PtsaN-C in primary cardiomyocytes (Supplementary Fig. 20 and Supplementary Movies 1–4).

**Fig. 6 | ROS scavenging and antiapoptotic effects of PtsaN-C on OGD/R-induced cell injury. a** Schematic diagram of the experimental approach. **b** Intracellular ROS detected by flow cytometry. **c**–**e** The levels of cellular MDA (**c**), SOD (**d**), and GSH (**e**) ($n = 6$ for biologically independent samples; **c** $^{ns}P_{(Unstim, PtsaN-C)} > 0.9999$, $^{***}P_{(Unstim, OGD/R)} < 0.0001$, $^{*}P_{(Unstim, OGD/R+PtsaN-C)} = 0.0094$, $^{***}P_{(OGD/R, OGD/R+PtsaN-C)} = 0.0008$; **d** $^{ns}P_{(Unstim, PtsaN-C)} > 0.9999$, $^{***}P_{(Unstim, OGD/R)} < 0.0001$, $^{***}P_{(Unstim, OGD/R+PtsaN-C)} = 0.0003$, $^{***}P_{(OGD/R, OGD/R+PtsaN-C)} < 0.0001$; **e**, $^{ns}P_{(Unstim, PtsaN-C)} > 0.9999$, $^{***}P_{(Unstim, OGD/R)} < 0.0001$, $^{**}P_{(Unstim, OGD/R+PtsaN-C)} < 0.0001$, $^{***}P_{(OGD/R, OGD/R+PtsaN-C)} < 0.0001$). **f** Representative fluorescent images of JC−1 and relative quantification of fluorescence ratio ($n = 6$ for biologically independent samples; $^{ns}P_{(Unstim, PtsaN-C)} > 0.9999$, $^{***}P_{(Unstim, OGD/R)} < 0.0001$, $^{***}P_{(Unstim, OGD/R+PtsaN-C)} = 0.0002$, $^{***}P_{(OGD/R, OGD/R+PtsaN-C)} < 0.0001$; scale bar: 50 μm). **g** Representative flow cytometry detection of cell apoptosis and relevant quantification of apoptotic cell ($n = 6$ for biologically independent samples; $^{ns}P_{(Unstim, PtsaN-C)} > 0.9999$, $^{***}P_{(Unstim, OGD/R)} < 0.0001$, $^{***}P_{(Unstim, OGD/R+PtsaN-C)} < 0.0001$, $^{***}P_{(OGD/R, OGD/R+PtsaN-C)} < 0.0001$). **h, i** Apoptosis of cells accessed by Tunel assay, with quantitative results (**h**) and representative images (**i**) ($n = 6$ for biologically independent samples; $^{ns}P_{(Unstim, PtsaN-C)} > 0.9999$, $^{***}P_{(Unstim, OGD/R)} < 0.0001$, $^{***}P_{(Unstim, OGD/R+PtsaN-C)} < 0.0001$, $^{***}P_{(OGD/R, OGD/R+PtsaN-C)} < 0.0001$; scale bar: 50 μm). **j, k** Immunofluorescence staining of phosphorylated γ-H2AX (**j**), and the proportion of positive nuclei (**k**) ($n = 6$ for biologically independent samples; $^{ns}P_{(Unstim, PtsaN-C)} > 0.9999$, $^{***}P_{(Unstim, OGD/R)} < 0.0001$, $^{***}P_{(Unstim, OGD/R+PtsaN-C)} < 0.0001$, $^{***}P_{(OGD/R, OGD/R+PtsaN-C)} < 0.0001$; scale bar: 50 μm). **l** Representative images and quantification of Western blot of C-casp3, Bax, Bcl-2 ($n = 6$ for biologically independent samples; for C-casp3 and Bax, $^{ns}P_{(Unstim, PtsaN-C)} > 0.9999$, $^{***}P_{(Unstim, OGD/R)} < 0.0001$, $^{***}P_{(Unstim, OGD/R+PtsaN-C)} < 0.0001$, $^{***}P_{(OGD/R, OGD/R+PtsaN-C)} < 0.0001$; Bcl-2, $^{ns}P_{(Unstim, PtsaN-C)} = 0.4165$, $^{***}P_{(Unstim, OGD/R)} < 0.0001$, $^{***}P_{(Unstim, OGD/R+PtsaN-C)} < 0.0001$, $^{***}P_{(OGD/R, OGD/R+PtsaN-C)} < 0.0001$). Data are analyzed using One-way ANOVA with Bonferroni post hoc test, and represented with mean ± SEM. ns, no significance; FC, fold-change. Source data are provided as a Source Data file.

Given the involvement of oxidative stress in the cell apoptotic process, we proceeded to investigate the antiapoptotic effects of PtsaN-C. Flow cytometric analysis revealed that the OGD/R procedure induced massive cardiac cell apoptosis, while treatment with PtsaN-C significantly reduced the apoptosis rate (Fig. 6g). Simultaneously, TdT-mediated dUTP nick-end labeling (Tunel) staining revealed that OGD/R caused substantial cell apoptosis, which was remarkably alleviated by the administration of PtsaN-C (Fig. 6h, i). Consistent with the above results, the live/dead cell staining assay presented that PtsaN-C effectively inhibited cell death induced by OGD/R (Supplementary Fig. 21). Additionally, considering that ROS can induce irreversible oxidative damage to DNA, we then assessed the phosphorylated H2AX (γ-H2AX), which served as a DNA damage-responsive sensor[39]. Immunofluorescence analysis suggested that the increase of γ-H2AX mediated by OGD/R was alleviated by PtsaN-C, which was attributed to the clearance of ROS by PtsaN-C (Fig. 6j, k).

Next, we analyzed the expression levels of the pro-oxidant gene (Nox2) and antioxidant gene (Ho-1). The results demonstrated that OGD/R caused a significant increase in Nox2 expression and a decrease in Ho-1 expression. However, treatment with PtsaN-C reversed these effects (Supplementary Fig. 22a, b). The pro-apoptotic proteins (C-casp3, Bax) and antiapoptotic protein (Bcl-2) are closely associated with cell survival. Western blot analysis verified that OGD/R induced upregulation of C-casp3 and Bax, while downregulation of Bcl-2. Notably, this phenotype was reversed by the administration of PtsaN-C (Fig. 6l). Furthermore, treatment with PtsaN-C also decreased the marker of cell damage (LDH) (Supplementary Fig. 22c). Overall, these results indicated that PtsaN-C effectively scavenged ROS and provided strong cryoprotection during OGD/R injury.

## In vivo therapeutic effects of PtsaN-C on myocardial I/R injury

Due to the remarkable ROS clearance and antiapoptosis performance of PtsaN-C in vitro, we further investigated its antioxidative and antiapoptotic effects on myocardial I/R injury. The myocardial I/R injury was applied with transient left anterior descending (LAD) occlusion for 45 min, followed by reperfusion for the indicated time (Fig. 7a). PtsaN-C was injected orthotopically when the ligation was loosened. After 24 h of reperfusion, both SOD and GPx activity were severely compromised, and the liquid peroxidation MDA content was significantly increased (Fig. 7b–d). However, treatment with PtsaN-C reversed these scenarios, indicating that PtsaN-C effectively decreased ROS levels during I/R injury. With respect to myocardial zymograms, used as indicators of myocardial injury, the PtsaN-C-treated group exhibited significantly lower levels of creatine kinase (CK), creatine kinase-MB (CK-MB), and lactate dehydrogenase 1 (LDH1) in the serum compared to I/R group (Fig. 7e–g). Moreover, PtsaN-C presented a greater therapeutic advantage over natural catalase, in terms of antioxidant effects, reducing cardiac damage and inflammation (Fig. 7h and Supplementary Fig. 23). Additionally, as reflected by 2,3,5-triphenyltetrazolium (TTC) and Evans blue staining, the I/R mice injected with PtsaN-C effectively reduced the area at risk (AAR) compared to the I/R group (Fig. 7i). These results indicated that PtsaN-C could minimize myocardium injury via ROS clearance after I/R injury.

To investigate the potential antiapoptotic function of PtsaN-C, we subsequently performed a series of apoptosis-related experiments. The quantitative real-time polymerase chain reaction (QRT-PCR) analysis demonstrated that the PtsaN-C group possessed a significantly higher mRNA expression level of Ho-1 and a lower level of Nox2 compared to the I/R group (Supplementary Fig. 24a, b). Tunel staining revealed that the massive apoptosis of cardiomyocytes induced by I/R injury was alleviated with the assistance of PtsaN-C (Fig. 7j). Additionally, the levels of pro-apoptotic proteins C-casp3 and Bax were upregulated, while Bcl-2 was downregulated during I/R injury (Fig. 7k). Administration of PtsaN-C not only reduced the expression of C-casp3 and Bax, but also increased the level of Bcl-2, indicating the capability of PtsaN-C in preventing apoptosis after I/R injury (Fig. 7k and Supplementary Fig. 24c–e).

Post-ischemic repair is a complex pathological process that involves cardiac fibrosis and remodeling as the disease progresses[40]. We also assessed the long-term effects of PtsaN-C on the prognosis of I/R injury. Functionally, we conducted dynamic monitoring of cardiac function over a series of time points. Echocardiography showed the cardiac function in mice treated with PtsaN-C was better than that of the I/R group from the first day after surgery until the end of the observation period, which was evidenced by higher ejection fraction (EF), higher fraction shortening (FS), smaller left ventricular internal diameters (LVID) and left ventricular volume (LV Vol) (Fig. 7l, Supplementary Fig. 25, Supplementary Table 8). These results were also supported by echocardiography with short-axis views of the left ventricle (Supplementary Fig. 26 and Supplementary Table 8). Histologically, in accordance with the previous results, administration of PtsaN-C significantly reduced the infarct size and border zone transition as assessed by the pipeline analysis of Picrosirius red staining images at day 14 post-injury, indicating that PtsaN-C treatment could alleviate infarct area and adverse cardiac remodeling (Fig. 7m and Supplementary Fig. 27a, b). Additionally, polarized light assay revealed a significant reduction in the fibrotic area after PtsaN-C treatment (Supplementary Fig. 27c). Cardiomyocyte loss and increased fibrosis contribute to hypertrophic response in extant cardiomyocytes[41]. Wheat germ agglutinin (WGA) staining reflected a marked decrease in cardiomyocyte area with PtsaN-C treatment after I/R injury (Fig. 7n and Supplementary Fig. 27d). Besides, all the results above also presented that PtsaN-C injection alone group showed no differences compared to the Sham group, indicating the favorable biocompatibility of PtsaN-C. Additionally, PtsaN-C possessed more efficiency for alleviating ROS-induced damage compared to PtnpN-C in vitro and in vivo, which again proved more efficient catalytic performance of PtsaN-C (Supplementary Fig. 28).

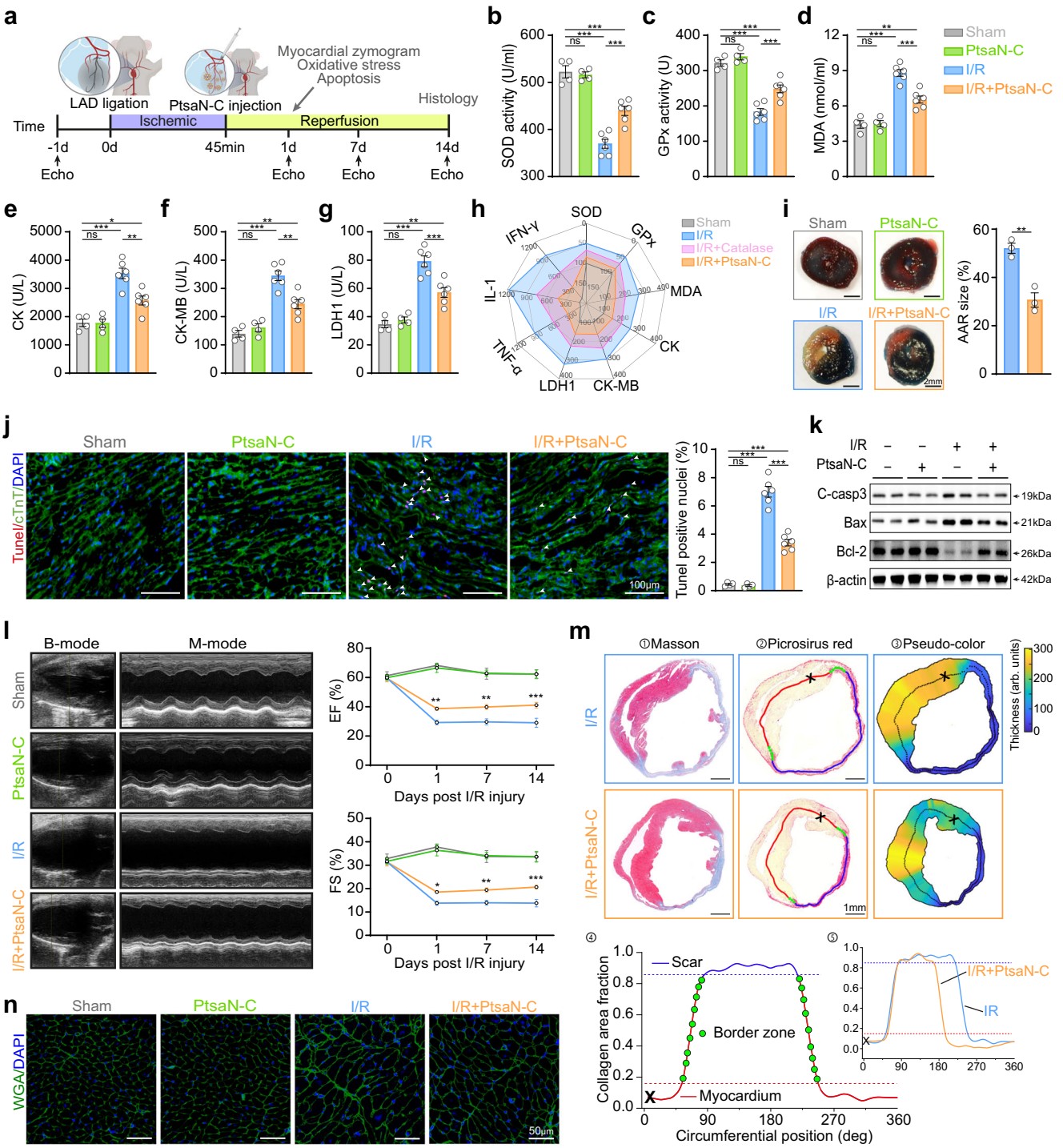

**Fig. 7 | Therapeutic efficacy of PtsaN-C on cardiac I/R injury. a** Schematic illustration of animal intervention. **b–d** The levels of SOD (**b**), GPx (**c**), MDA (**d**) in cardiac tissue homogenate (**b**, $^{ns}P_{(Sham, PtsaN-C)} > 0.9999$, $^{***}P_{(Sham, I/R)} < 0.0001$, $^{***}P_{(Sham, I/R+PtsaN-C)} = 0.0002$, $^{***}P_{(I/R, I/R+PtsaN-C)} = 0.0003$; **c**, $^{ns}P_{(Sham, PtsaN-C)} > 0.9999$, $^{***}P_{(Sham, I/R)} < 0.0001$, $^{***}P_{(Sham, I/R+PtsaN-C)} = 0.0008$, $^{***}P_{(I/R, I/R+PtsaN-C)} = 0.0008$; **d**, $^{ns}P_{(Sham, PtsaN-C)} > 0.9999$, $^{***}P_{(Sham, I/R)} < 0.0001$, $^{**}P_{(Sham, I/R+PtsaN-C)} = 0.0010$, $^{***}P_{(I/R, I/R+PtsaN-C)} = 0.0002$). **e–g** Serum concentrations of myocardial enzyme spectrum CK (**e**), CK-MB (**f**), and LDH1 (**g**) (**e**, $^{ns}P_{(Sham, PtsaN-C)} > 0.9999$, $^{***}P_{(Sham, I/R)} < 0.0001$, $^{*}P_{(Sham, I/R+PtsaN-C)} = 0.0294$, $^{**}P_{(I/R, I/R+PtsaN-C)} = 0.0025$; **f**, $^{ns}P_{(Sham, PtsaN-C)} > 0.9999$, $^{***}P_{(Sham, I/R)} < 0.0001$, $^{**}P_{(Sham, I/R+PtsaN-C)} = 0.0022$, $^{**}P_{(I/R, I/R+PtsaN-C)} = 0.0011$; **g**, $^{ns}P_{(Sham, PtsaN-C)} > 0.9999$, $^{***}P_{(Sham, I/R)} < 0.0001$, $^{**}P_{(Sham, I/R+PtsaN-C)} = 0.0017$, $^{***}P_{(I/R, I/R+PtsaN-C)} = 0.0007$). **h** Radar chart plot of the indicated indexes. **i** Representative photographs of Evans blue/TTC double staining and the corresponding quantified results (*n* = 3 for biologically independent animals; $^{**}P = 0.0042$; scale bar: 2 mm). **j** Representative image of apoptotic cells evaluated by Tunel assay ($^{ns}P_{(Sham, PtsaN-C)} > 0.9999$, $^{***}P_{(Sham, I/R)} < 0.0001$, $^{***}P_{(Sham, I/R+PtsaN-C)} < 0.0001$, $^{***}P_{(I/R, I/R+PtsaN-C)} < 0.0001$; scale bar: 100 μm). **k** Representative western blot of indicated proteins. **l** Cardiac function detected by echocardiography (comparison between I/R and I/R + PtsaN-C; EF: $^{**}P_{(1d)} = 0.0046$, $^{**}P_{(7d)} = 0.0018$, $^{***}P_{(14d)} = 0.0002$; FS: $^{*}P_{(1d)} = 0.0339$, $^{**}P_{(7d)} = 0.0096$, $^{***}P_{(14d)} = 0.0005$). **m** Pipeline analysis of scar thickness, infarct size, and border zone transition (scale bar: 1 mm). **n** Representative WGA staining of cardiomyocytes area in border zone (scale bar: 50 μm). Data are represented with mean ± SEM. In **b–g** and **j**, *n* = 4 biologically independent animals for Sham groups and 6 for I/R groups. **b–g** and **j** are analyzed with One-way ANOVA with Bonferroni post hoc test, **i** with two-tailed unpaired Student *t*-test, and **l** with Two-way ANOVA with Bonferroni post hoc test. Echo, echocardiography; ns, no significance. Source data are provided as a Source Data file.

In summary, these in vivo results illustrated that PtsaN-C possessed distinct therapeutic effects on antioxidant and antiapoptosis in mice with myocardial I/R injury.

## Therapeutic mechanisms of PtsaN-C on myocardial I/R injury

To further investigate the cardiac protective mechanisms of PtsaN-C in I/R circumstances, we performed a label-free proteomics analysis. Heart tissue from the injured region was pooled after 24 h of reperfusion, and then subjected to protein extraction, digestion, and liquid chromatography-mass spectrometry/mass spectrometry (LC-MS/MS) detection (Fig. 8a). Both partial least squares discrimination analysis (PLS-DA) and orthogonal projections to latent structures discriminant analysis (OPLS-DA) demonstrated high similarity between the Sham and PtsaN-C groups, indicating the biosafety for PtsaN-C administration (Fig. 8b and Supplementary Fig. 29a). Conversely, clusters between I/R and I/R PtsaN-C groups were separated, highlighting divergent protein expression profiles. Volcano plots showed significantly differentially expressed proteins (DEPs) between I/R and I/R PtsaN-C groups (Fig. 8c). Additional pairwise comparisons were shown (Supplementary Fig. 29b). Subsequently, an in-depth analysis of the DEPs was performed to investigate whether PtsaN-C could induce the expression of proteins associated with cardioprotection in the context of I/R. A heatmap illustrated that administration of PtsaN-C altered the protein profiles associated with MAPK signaling, calcium homeostasis, apoptosis, energy metabolism, response to oxidative stress, and oxidation-redox homeostasis, which have been reported closely related to cardioprotection in the context of I/R[42] (Fig. 8d). Furthermore, protein-protein interaction network analysis of the DEPs verified strong correlations among the six protein clusters, indicating that PtsaN-C exerted its protective functions through multiple mechanisms (Fig. 8e). Gene set enrichment analysis (GSEA) indicated that the intrinsic apoptotic signaling pathway in response to oxidative stress and the Tnf-α induced inflammatory response were significantly enriched in the I/R group compared to the I/R PtsaN-C group (Fig. 8f and Supplementary Fig. 30a). These results reflected that PtsaN-C exerted its protective functions through multiple and intricate pathways.

To elucidate the key pathway, gene ontology (GO) analysis was performed among the DEPs. Enrichment analysis of cellular components revealed that the DEPs were predominantly localized in the cytoplasm, endosome, vesicle, and mitochondria (Supplementary Fig. 30b). The major molecular function of these DEPs was mainly related to catalytic and binding activity (Supplementary Fig. 30c). Furthermore, the biological process of the DEPs encompassed the apoptotic process, oxidation-reduction process, mitochondrial respiration, and inflammatory response, which played important roles in the pathophysiology of I/R injury (Fig. 8g). To delve deeper into the specific signaling pathway modulated by PtsaN-C, we performed an ingenuity pathway analysis (IPA). The results predicted that PtsaN-C predominantly depressed the activation of the MAPK/Jnk pathway (Fig. 8h). It has been widely reported that the inhibition of MAPK/Jnk exerts cardioprotective properties[43]. Consistent with the IPA analysis, western blot demonstrated that the content of phospho-Jnk (p-Jnk) reduced after PtsaN-C intervention (Supplementary Fig. 31a). Immunofluorescent staining also revealed that the nuclear translocation of Jnk was mitigated after PtsaN-C administration (Supplementary Fig. 31b). These results illustrated that PtsaN-C exerted its cardioprotective effects through multiple mechanisms, including ROS clearance, maintenance of calcium and redox homeostasis, and suppression of MAPK/Jnk signaling pathway, thereby preventing cell injury induced by I/R stimuli (Fig. 8i).

## In vivo biosafety assessment of PtsaN-C

Biosafety and toxicity must be evaluated for the clinical application of novel nanomedicine. Therefore, the biocompatibility of the PtsaN-C was assessed systematically. Previous in vitro and in vivo experiments have partially verified the good biocompatibility of PtsaN-C. Further proteomic analysis, comparing the Sham and PtsaN-C groups revealed that the sole administration of PtsaN-C could enhance the oxidoreductase activity, binding activity, disulfide oxidoreductase activity, and ATPase activity, which could further engage in biological processes pertaining to cell redox homeostasis, inflammation regulation, DNA repair, and hypoxia response (Fig. 9a, b and Supplementary Fig. 32). This indicated that PtsaN-C treatment played similar functions to that of ischemic preconditioning treatments, which were recognized to mitigate I/R-induced cardiac injuries[42]. Given that the hemolysis rate of nanomedicines must be less than 10% for systemic application[44], our results showed a very low incidence of hemolysis even at concentrations of up to 1000 μg/ml, which was far higher than the dosages employed in vivo (Supplementary Fig. 33). Furthermore, there was no evidence of necrosis, inflammatory infiltration, hemorrhage, or fibrosis from the histological examinations of the major organs (Fig. 9c).

Previous studies have verified that quantum dots can be degraded in vivo, and diameters smaller than 20 nm can be eliminated through renal excretion, while larger sizes are excreted via hepatobiliary clearance[45–47]. The distribution of PtsaN-C was determined by measuring the concentration of Pt via inductively coupled plasma-mass spectrometry (ICP-MS). Our result revealed that PtsaN-C mainly accumulated in the heart, with partial accumulation observed in the liver and kidney (Fig. 9d, e). The amount of PtsaN-C decreased in the heart over time, while the rate of decline was much slower in the liver and kidney, suggesting the accumulation, degradation, and excretion of PtsaN-C in these organs. Coincident with the accumulation trend in the kidney, the content of Pt also increased in urine, indicating potential kidney excretion (Supplementary Fig. 34). Besides, PtsaN-C was negligibly distributed in the spleen without causing tissue injury (Supplementary Fig. 35).

We also detected the inflammatory cytokine in serum. The levels of TNF-α and IL-6 showed no significant difference between the Sham and PtsaN-C groups, demonstrating that PtsaN-C wouldn't induce an inflammatory response (Fig. 9f). Additionally, the liver and kidney function indexes in the PtsaN-C group were identical to the Sham group, indicating that PtsaN-C wouldn't cause liver and kidney injury (Fig. 9g, h and Supplementary Fig. 36a). Moreover, the good biosafety was further confirmed by complete blood cell analysis which showed no difference among all blood cells (Fig. 9i and Supplementary Fig. 36a, b). All these results together verified the biosafety and excellent biocompatibility of PtsaN-C.

## Discussion

In this study, we successfully designed an artificial single-atom nanozyme (SAN) with a structure of Pt-N$_4$-C to imitate natural antioxidant defense system for efficient ROS clearance and cytoprotection (Fig. 10). Compared to the PtnpN-C structure, PtsaN-C exhibited exceptional catalytic efficacy in mimicking SOD, CAT, POD, and GPx-like properties. This performance can be attributed to the synergistic effect and metallic electronic structure of the catalytic center. Additionally, the Pt-N$_4$-C structure maximized the efficiency of atomic utilization. Both in vitro and in vivo studies unequivocally demonstrated that PtsaN-C not only effectively scavenged ROS to protect cells from apoptosis under I/R-induced pathological conditions, but also exhibited excellent biocompatibility and biosafety. Mechanistically, PtsaN-C exerted cardioprotective effects through multiple pathways, including the regulation of oxidation-redox homeostasis, calcium homeostasis, energy metabolism, apoptosis, and the potential modulation of the MAPK/Jnk signaling pathway.

Numerous traditional antioxidant drugs have been developed for the treatment of myocardial I/R injury, including carvedilol, baicalin, and coenzyme Q10, while their clinical efficacy is limited due to poor bioavailability and obvious side effects[21,48]. Since the Fe$_3$O$_4$

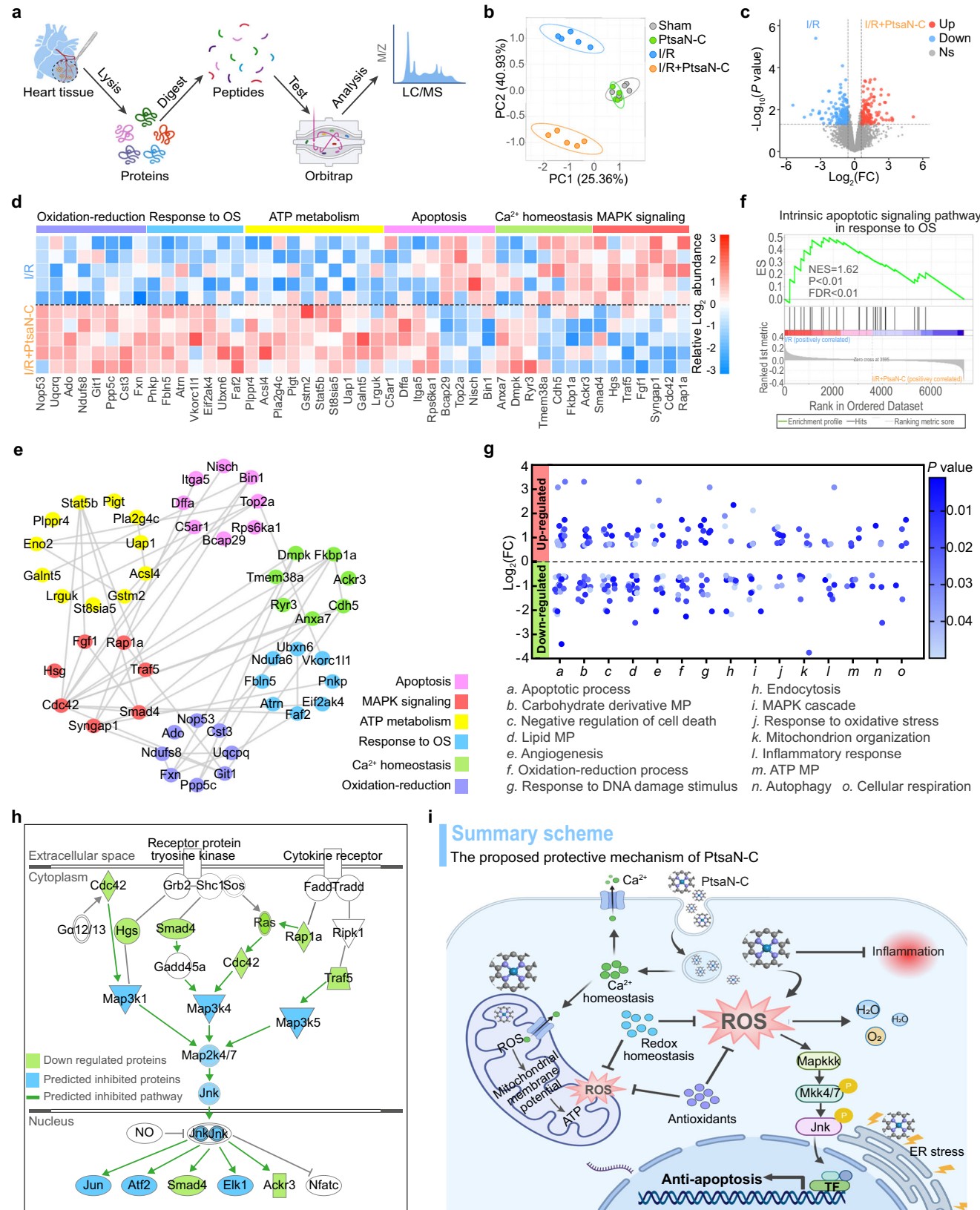

nanoparticles have been identified as ROS scavengers, various nanozymes have garnered significant interest[49]. Among antioxidant nanozymes, metal/metal oxide nanozymes, represented by cerium oxide and iron oxide, have been employed to treat myocardial ischemia-reperfusion injury[22,50]. Although these nanozymes exert ROS-scavenging properties and protect the myocardium from oxidative injury, their side effects hinder their bioapplication. Ceria nanozymes can induce inflammatory responses to damage vascular structure and aggravate myocardial I/R injury[51]. Iron oxide nanozymes can cause iron overload, leading to oxidative stress and exacerbating myocardial damage[52]. Recently, PtNPs, verified as highly efficient ROS scavengers with multienzyme-mimicking properties, have been demonstrated to

**Fig. 8 | Mechanistic explanation of the therapeutic effects of PtsaN-C on cardiac I/R injury. a** Schematic representation of label-free proteomic workflow. **b** Partial least squares discrimination analysis of the samples. **c** Volcano plots comparing protein levels between I/R and I/R PtsaN-C groups (Significantly differentially expressed proteins were defined with |fold-change | > 1.5 and $p < 0.05$). **d** Heatmap illustrating the differentially expressed proteins enriched in I/R and I/R PtsaN-C groups, probably exerting cardioprotection by mitigating I/R-associated pathological processes. **e** Protein-protein interaction network of the proteins possibly involved in I/R-induced pathological processes. **f** GSEA enrichment analysis for

differentially expressed proteins ($P < 0.001$). **g** Dominant biological processes associated with the response to I/R. **h** Ingenuity pathway analysis of predicting upstream regulator among differentially expressed proteins. **i** Schematic illustration of potential cardioprotective mechanisms mediated by PtsaN-C. Data are processed with a moderated two-tailed t-test (Limma) for **c**, **g**; an empirical phenotype-based permutation test for **f**. OS, oxidative stress; NSE, normalized enrichment scores; MP, metabolism process; TF, transcription factor; ER, endoplasmic reticulum. Source data are provided as a Source Data file.

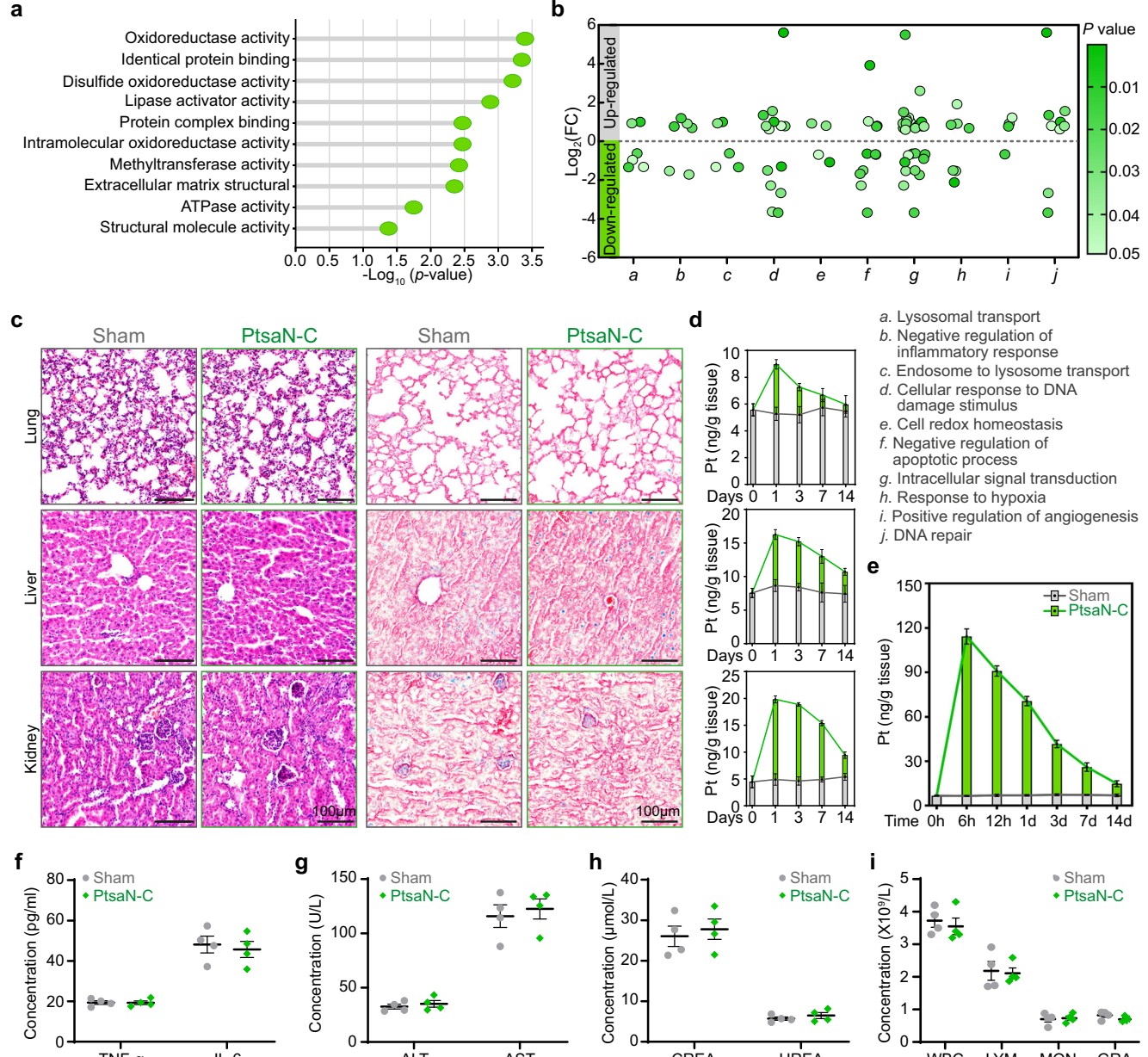

**Fig. 9 | In vivo biosafety of PtsaN-C. a**, **b** Molecular function (**a**) and biological processes (**b**) enrichment analysis of differentially expressed proteins between the Sham and PtsaN-C groups. **c** Evaluation of the in vivo toxicity of PtsaN-C to major organs through histological analysis of hematoxylin-eosin (HE) and Masson trichrome staining at day 14 post-administration. Experiments were repeated three times with similar results (scale bar: 100 μm). **d** ICP-MS analysis of lung (upper), liver (middle), and kidney (down) at indicated time points ($n = 3$ for biologically independent animals). **e** The concentration of Pt in heart tissue at various time

points after PtsaN-C injection ($n = 3$ for biologically independent animals). **f**–**h** Assessment of serum inflammatory factors TNF-α and IL-6 (**f**), liver function indexes ALT and AST (**g**), renal function indexes CREA and UREA (**h**) at 24 h after intervention ($n = 4$ for biologically independent animals). **i** Examination of blood parameters in mice at 24 h post-injection ($n = 4$ for biologically independent animals). Data are presented with mean ± SEM. Data are analyzed with two-tailed Fisher's exact test for **a**; moderated two-tailed t-test (Limma) for **b**; and two-tailed unpaired Student t-test for **f**–**i**. Source data are provided as a Source Data file.

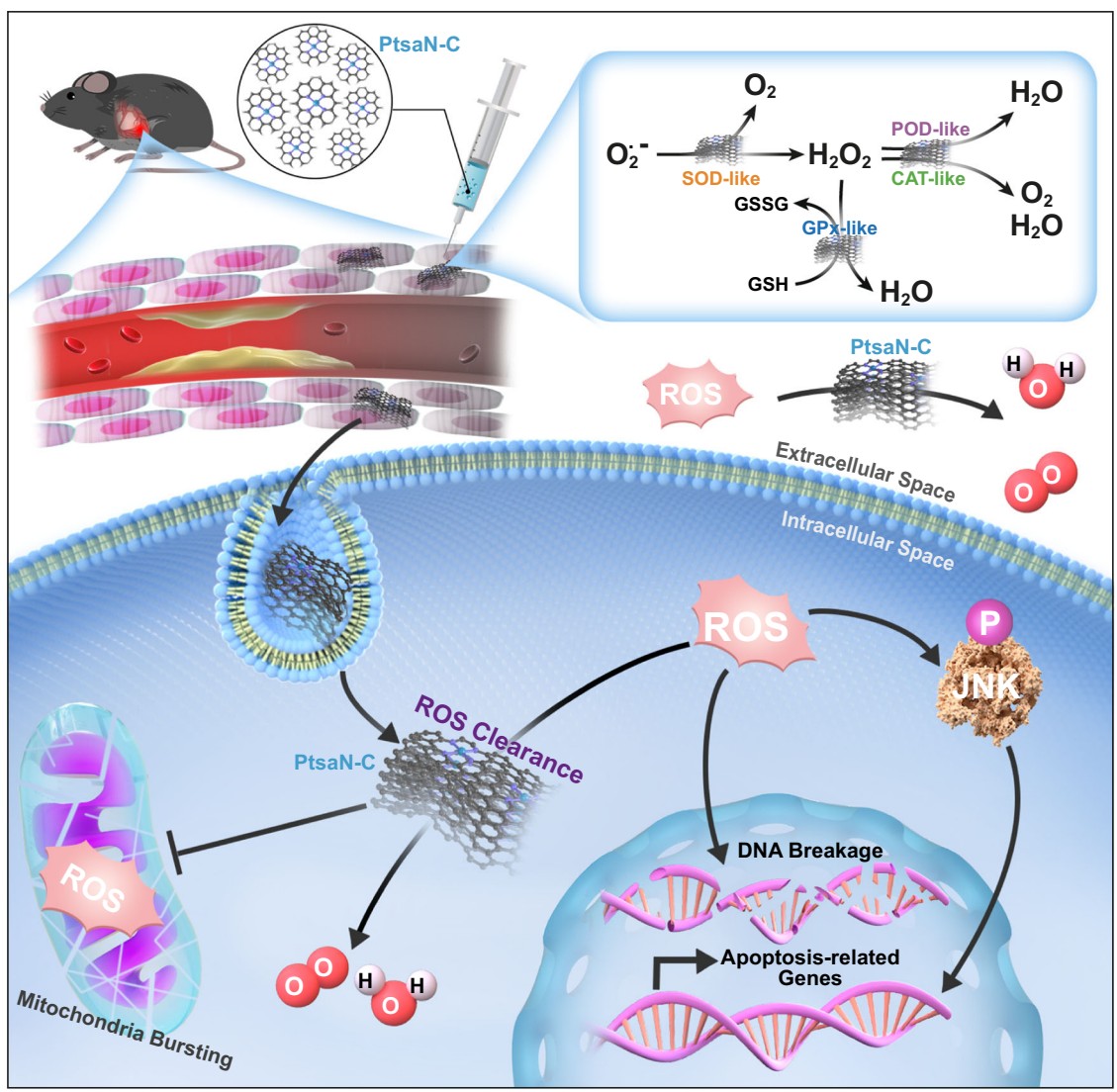

**Fig. 10 | Schematic illustration of the protective effects of PtsaN-C on myocardial ischemia/reperfusion injury.**

safely alleviate I/R-induced hepatic, renal, and neuronal injury[28–30,32,33,53]. Besides, PtNPs can imitate the mitochondrial complex I NADH ubiquinone oxidoreductase activities, and quinone oxidase-like activity as well[33]. Moreover, high chemical stability and resistance to aggregation confer PtNPs the advantage of coping with diverse and complex physiological environments. However, literature has revealed that the catalytic activity of Pt is inversely related to the particle diameter, hinting the smaller, the better[32,54].

Compared to traditional nanozymes, SANs possess distinct efficiency of catalytic performance, owing to fully exposed atomic active metal sites, almost 100% atom dispersion, and full atomic utilization[55,56]. The metal-nitrogen coordination centers in SANs induce or tailor charge transfer effects, significantly boosting the inherent activity of the active site for providing highly efficient catalysis[56]. In addition, this unique electronic/geometrical structure theoretically possesses enormous potential to mimic the structure and catalytic performance of natural metalloenzymes. Besides, the well-defined active sites offer a promising model to illustrate the mechanism of structure-activity relationship[55,56]. Given these, we propose the hypothesis that designing Pt single-atom nanozymes, with a Pt-N$_4$-C (PtsaN-C) structure, can exhibit excellent overall catalytic functionality.

In fact, PtsaN-C not only inherits the advantages of PtNPs with respect to multienzyme mimicry activity and high stability, but also leverages the unique strength of SANs with extraordinary catalytic efficiency and enzyme-like selectivity. When compared with recently reported metal-based antioxidative nanozymes, PtsaN-C exhibits a highest catalytic efficiency in terms of Km and TON. Thus, PtsaN-C has more advantageous applications in complex biological systems. Our in vivo results validate the significant therapeutic effects of PtsaN-C in mitigating oxidative stress and apoptosis in mouse myocardial I/R injury. Additionally, histological and serum examinations confirm the absence of cytotoxicity at therapeutic doses of PtsaN-C. Overall, PtsaN-C efficiently eliminates ROS induced by I/R injury, restores intracellular homeostasis, prevents apoptosis, and significantly reduces the biotoxicity of conventional platinum nanozymes. Considering both remarkable catalytic performance and therapeutic effects, we believe that PtsaN-C possesses the following advantages over other nanomaterials in I/R injury bioapplication: (I) High catalytic efficiency. The fully exposed Pt-N$_4$-C coordination centers induce charge transfer to boost and amplify the catalytic efficiency of Pt; (II) High stability. PtsaN-C not only has the inherent stability of Pt, but also inherits the advantages of heterogeneous catalysts, making PtsaN-C more efficient in diverse and complex physiological environments; (III) High biocompatibility.

PtsaN-C provides nearly 100% atomic dispersion and maximizes Pt utilization, which reduces metal content to a large extent to minimize toxicity.

However, there exists some lack in our study. PtsaN-C was administrated via intramyocardial injection which was more difficult to manipulate compared to intravenous administration during clinical translation. Enhancing the cardiac targeting via intravenous treatment may be achieved by appropriate modification of PtsaN-C in a later study. Our study was performed based on a mouse model. However, it should be noted the differences between mouse models and patients. Further validation experiments should be carried out in large animals before the clinical application of PtsaN-C.

In this work, we present PtsaN-C as a nanozyme for ROS clearance with high catalytic efficiency and biosafety, which provides a promising therapeutic option for the treatment of myocardial I/R injury.

# Methods

### Animals
All animal studies were conducted in strict compliance with the National Institutes of Health (NIH) guidelines and approved by the Animal Care and Use Committee of Shanghai Jiao Tong University (approval number: O_A2023071). *C57BL/6C* mice (6–8 weeks old, male) and newborn *Sprague-Dawley* rats (1 to 3 days after birth) were obtained from SLRC Laboratory Animal Co., Ltd. (Shanghai, China). The *C57BL/6C* mice were bred in a specific pathogen-free environment. The mice were housed under standard conditions with a 12/12 h light-dark cycle and given ad libitum access to water and food. The room was kept at a temperature range of 20–25 °C, with a relative humidity maintained between 40% and 70%. The mice were euthanized by inhaling carbon dioxide followed by dislocating their cervical spines. To avoid the effect of sex hormones secreted by female mice on the results, only male *C57BL/6C* mice were included in the study. The newborn *Sprague-Dawley* rats were included without discrimination of sex, during isolation of neonatal rat cardiomyocytes.

### Synthesis of catalysts
Firstly, a solution of $Na_2PtCl_4$ (Adamas, >99%) was prepared by dissolving it in 200 ml of Millipore water (18.2 MΩ·cm) to achieve a concentration of 5 mg/ml. Then, 0.5 ml of this solution was combined with 40 ml of GQDs (XFNano) solution in a centrifuge tube, followed by ultrasonication for 30 min. Afterward, the sample in the centrifuge tube was freeze-dried and then thoroughly mixed with urea (VWR, >99%) at a mass ratio of 1:10 through grinding for 15 min. The mixture was subsequently placed in a tube furnace and calcined at a temperature of 500 °C under a gas flow of 100 sccm Ar (UHP, Airgas) within 2 h, resulting in the formation of a compound named PtsaN-C. Expanding on this procedure, the atmosphere was changed to a 10 vol % $H_2$/Ar mixture (UHP, Airgas), and the sample was calcined at 600 °C for 1 h to yield a product named PtnpN-C. The synthesis of the N-C reference catalyst followed the same procedure as that of the PtsaN-C catalyst, with the sole exception being the omission of the Pt stock solution in the GQDs solution.

### Transmission electron microscopy (TEM) test
TEM was conducted on a Tecnai G2 F20 S-Twin transmission electron microscope operating at 200 kV. The morphology of atomic dispersion was observed using a high-angle annular dark field scanning TEM (HAADF-STEM) with two aberration correctors (ThermoFisher Themis Z microscope). Energy dispersive X-ray spectroscopy (EDS) was conducted using four-column Super-X detectors. The adsorption and desorption of $N_2$ were measured at 77 K liquid nitrogen atmosphere, using a 4-station automatic specific surface area analyzer model APSP-2460 (Micromeritics). The BET method was employed to calculate the total BET-specific surface area of the material.

### High-resolution X-ray diffraction (XRD) analysis
With a scan rate of 0.02° per step and 4 s per step, the XRD patterns were monitored using a Rigaku Mini Flex 600 spectrometer.

Based on the Rietveld structural refinement framework which was implemented in the General Structure Analysis System (GSAS, version 1.0) package, the XRD patterns were analyzed. Kratos AXIS Ultra DLD XPS instrument was used to acquire XPS and UPS data using the parameter of Al-kα source ($hv = 1486.68$ eV) and He I ($hv = 21.22$ eV) illumination. XPS binding energy was calibrated to 284.8 eV using adventitious alkyl carbon C $1s$ peak. Inductively coupled plasma optical emission spectrometry (ICP-OES, ThermoFisher, iCAP 6300) was used to determine the concentration of Pt loading. X-ray absorption spectroscopy (XAS) at Pt L3-edge was measured at the beamline BL14W1 of Shanghai Synchrotron Radiation Facility (XAFS station, SSRF). Demeter (version 0.9.26) software packages were used to process and analyze XANES and EXAFS data.

### In situ enhanced Raman spectra analysis
Raman spectra were collected through a confocal XploRA Raman spectrometer (XploRA, France) with a 638 nm laser (0.15 mW) and 50 × microscope objective lens (numerical aperture = 0.55). The PtsaN-C and PtnpN-C, loaded onto commercial Au@SiO2 nanoparticles, were introduced into the culture medium containing H9C2 cells line (at a density of 70%–80%) before SERS measurements respectively. 100 ng/ml of Phorbol myristate acetate (PMA, Sigma, >99%) was utilized to facilitate ROS generation. PtsaN-C and PtnpN-C were added into the culture medium respectively before PMA stimulation.

### SOD-like activity of PtsaN-C
The SOD-like activity was detected with SOD Activity Assay Kit (Solarbio) in accordance with the manufacturer's instructions. Briefly, the working solution was incubated with PtsaN-C (5 μg/ml) or equal doses of references, then absorbance changes were monitored via UV-vis-NIR Epoch microplate reader at 560 nm.

### CAT-like activity of PtsaN-C
The CAT-like activity was investigated via two methods. One method was using TPA which can react with ·OH to switch to a fluorescence aromatic hydroxylated product (TAOH). Hence, the CAT-like activity was measured by detecting the fluorescent signal. 10 mM $H_2O_2$ (Sigma, 35%) solution was mixed with catalase (5 U/ml, Sigma), PtsaN-C (5 μg/ml), and equal doses of references respectively for reaction, then TPA (Ourchem, 99%) was added to the final concentration of 500 μM[36]. The fluorescent signal was monitored via fluorescence spectroscopy SpectraMax iD3. Another method was accessed based on the $O_2$ generation decomposed from $H_2O_2$ by using a dissolved oxygen meter. Dissolved $O_2$ in various concentrations of $H_2O_2$ mixed with indicated catalysts (5 μg/ml), respectively, were monitored at time series. Michaelis–Menten kinetic analysis referred to maximum reaction velocity ($V_{max}$) and Michaelis–Menten constant ($K_m$) were performed via the Lineweaver–Burk equation by the GraphPad Prism software (version 9.3.0).

### POD-like activity of PtsaN-C
The POD-like activity of PtsaN-C was performed with TMB (Solarbio, 99.5%) and $H_2O_2$ as substrates[36]. The HAc/NaAc buffer (0.1 M, pH 4.5) mixed with TMB (0.8 Mm), PtsaN-C (5 μg/ml), and different concentrations of $H_2O_2$ were reacted at room temperature. With the formation of oxTMB, the solution turned blue, and absorbance at 652 nm was monitored. Subsequently, Vmax and Km were accessed by using Lineweaver–Burk equation.

### GPx-like activity of PtsaN-C
The GPx-like activity was accessed by using the GPx Activity Assay Kit (Beyotime) per the manufacturer's instructions, based on the couple

reductase method. In brief, PtsaN-C (5 µg/ml) or equal doses of references were performed to determine GPx-like activity. As GSH can be oxidized to GSSG, which can be in turn reduced to GSH by GR with NADPH, the amount consumption of NADPH can be measured at 340 nm via Epoch microplate reader to reveal the GPx-like property.

### •OH scavenging property of PtsaN-C
The EPR spectra were obtained using a Bruker A300 spectrometer. EPR signal was obtained as soon as $H_2O_2$ (5 mM), $FeSO_4$ (2 mM, Sigma, >99%), DMPO (0.1 M, Dojindo, >99%) and PtsaN-C (5 µg/ml) or equal doses of PtnpN-C (5 µg/ml) were added to HAc/NaAc buffer (0.1 M, pH 4.5).

### Density functional theory (DFT) calculation
First-principles calculations were conducted using the Vienna ab initio simulation package 6.1.0[57], based on the generalized gradient approximation proposed by Perdew, Burke, and Ernzerhof et al.[58]. The dispersion correction process adopts Grimme D3 correction[59]. A plane wave truncation energy of 450 eV was utilized. The energy convergence standard was set to $10^{-5}$ eV based on the Kohn-Sham equation[60]. The convergence standard for atomic force was 0.03 eV Å$^{-1}$. Brillouin zone integration was performed using a $2 \times 2 \times 1$ k-mesh[61]. The vacuum layer was set to 15 nanometers to reduce the interaction between periodic plates. The calculation of the transition state was carried out using the climbing-image nudged elastic band (cNEB) method based on the VTST plugin[62], which was further confirmed through frequency analysis. The calculation of Gibbs free energy is as follows[63]:

$$\Delta G = \Delta E + ZPE - T\Delta S$$

Where $\Delta E$ is the DFT energy of the model, ZPE is the zero-point energy, and $T$ is the temperature (298.15 K), $\Delta S$ is the entropy of the structure given by the vibration frequency. In addition, the calculated data was processed using VASPKIT code (VASP, version 6.1.0)[64]. Perform crystal orbital Hamiltonian population (COHP) analysis using LOBSTER package[65].

### Cell culture
Rat H9C2 cardiomyocyte cells were purchased from the American Tissue Culture Collection (ATCC, CRL-1446), and grown in DMEM medium (Gibco) with 10% FBS (Gibco) and 1% penicillin/streptomycin (Gibco) under a humidified atmosphere with 5% $CO_2$ at 37 °C.

### Neonatal rat cardiomyocytes (NRCMs) isolation
NRCMs were isolated from the *Sprague-Dawley rat* (1 to 3 days after birth). The ventricular tissues were harvested, sheared into small fragments, and then digested with 0.1% Trypsin/EDTA (TE, Gibco) at 37 °C for 20 min. Repeat digestion was conducted until complete digestion. Each round supernatant was collected and centrifuged at $300 \times g$ for 5 min at room temperature. The cells were resuspended with a complete DMEM medium and incubated for 2 h. Cardiomyocytes in suspension were seeded on gelatin-coated culture plates. The cells were cultured for 24–48 h before further experiments with 0.1 mM 5-Bromo-2-deoxyuridine (BrdU, Sigma, >98%).

### Ischemia/reperfusion model
Myocardial Ischemia/reperfusion was performed as previously described[66]. Briefly, mice were anesthetized with 1.5% isoflurane and intubated for ventilation at a rate of 120–130 strokes/min using a Harvard mini-vent ventilator. A thoracotomy was performed to visualize the heart, and then the left anterior descending coronary artery (LAD) was temporally ligated with an 8-0 suture for 45 min, followed by suture removal for reperfusion. During the release of the suture, local injection of PtsaN-C (500 µg/ml, 10 µl) on the injury area was conducted with a 32 G needle. For the sham procedure, mice were treated with a suture passed under the LAD without ligation and injected with an equal amount of saline after 45 min. Surgical wounds were sutured using sterile 5-0 sutures.

### In vitro oxygen–glucose deprivation and reperfusion (OGD/R) process and PtsaN-C intervention
The OGD/R model was performed according to the reported protocol[67]. In short, when the density of the cells reached 70%–80%, the H9C2 cells were exposed to 0.1% oxygen in serum-free no glucose DMEM at 37 °C to achieve oxygen–glucose deprivation for 8 h. Following incubation for 8 h, the media was replaced with complete growth media containing PtsaN-C (10 µg/ml) or similar dose of vehicle, and then cells were placed in a normoxic incubator for recovery 12 h.

### Cell viability assay
H9C2 cells were seeded at a density of 5000 cells per well into 96-well plates and incubated for 24 h. The medium was replaced with 100 µl fresh medium with various concentrations of PtsaN-C or commercial Pt nanoparticles (Merck). Following normoxic incubation for 24 h or OGD/R process, cells were rinsed with phosphate-buffered saline (PBS, Gibco, pH 7.4) twice. Then the CCK-8 assay (Yeasen) was performed according to the product instruction and monitored the absorbance at 450 nm.

### SOD, MDA, GSH, and ATP assays
The levels of SOD, MDA, GSH, and ATP were tested with SOD Activity Assay Kit (Solarbio), MDA Content Assay Kit (Solarbio), GSH Content Assay Kit (Solarbio), and ATP Content Assay Kit (Beyotime) respectively per manufacturer's instructions. Cells suffered OGD/R process or cardiac tissues with different disposes were collected and carried out in accordance with the manufacturer's protocol.

### ROS assay
ROS generation in H9C2 with different disposes was tested using a ROS Assay Kit (Beyotime). Cells were incubated with the fluorescence probe (DCFH-DA) at 37 °C for 20 min. Then the fluorescence was measured using a flow cytometer or fluorescent microplate reader with an excitation wavelength of 488 nm and an emission wavelength of 525 nm. The gating strategy was presented in Supplementary Fig. 37a.

### LDH assay
As apoptosis or dead cells released stable LDH, LDH release analysis was performed to assess cell injury. Cells after the OGD/R process were lysed and detected using the LDH Cytotoxicity Assay Kit (Yeasen) according to the kit instructions.

### Mitochondrial membrane potential
The mitochondrial membrane potential was assessed by JC-1 dye (Beyotime) per the manufacturer's instructions. Briefly, JC-1 dye presented potential-dependent accumulation in the mitochondria, with the fluorescence shift from red to green. Changes in mitochondrial membrane potential were calculated as the ratio between red and green fluorescence intensities. 1 ml JC-1 working solution was added to the cells with OGD/R disposes and incubated at 37 °C for 30 min. Cells were then immediately observed via fluorescence microscopy.

### Live/dead staining
LIVE/DEAD stain (ThermoFisher) was used for live/dead staining according to the manufacturer's description. After incubation with the dyes for 20 min at room temperature in the dark, images were captured on fluorescence microscopy.

## Echocardiography

Blinded echocardiographic measurements were performed on Vevo 2100. Mice were anesthetized and sustained with 1.5% isoflurane. Hearts were visualized from short-axis and long-axis views. B-mode and M-mode images were captured. The measurement of cardiac indexes was calculated automatically by echocardiographic built-in software.

## Myocardial enzyme spectrum

At 24 h after surgery, mouse blood was collected and centrifuged at $2500 \times g$ for 10 min at 4 °C to obtain serum. The myocardial enzyme spectrum was detected by an automatic biochemical analyzer.

## Hepatic and renal function

Blood was collected, anticoagulated with heparin, and centrifuged at $2500 \times g$ for 10 min at 4 °C to obtain serum. The hepatic and renal function was measured by an automatic biochemical analyzer.

## Blood routine examination

For blood routine examination, blood was collected from the orbital vein and mixed with EDTA immediately. The complete blood count was performed using an animal blood analyzer.

## Evans blue/TTC staining

The hearts were perfused with PBS to remove the blood remaining and perfused with 1% Evans blue (Sigma, > 75%). The hearts were then cut into 1-mm-thick cross sections which were incubated with 2% TTC (Sigma, > 98%) solution at 37 °C for 15 min in the dark. After that, the slices were fixed with 4% paraformaldehyde, and photographs of the slices were taken.

## Histology

Tissues were perfused with PBS to remove the blood and fixed with 4% formaldehyde for paraffin embedding and sectioning. Tissue sections were stained with Masson's trichrome Kit (Sigma) and Picrosirius red (Sigma, 25%) to identify the healthy regions and fibrotic regions per the manufacturer's guidelines. Picrosirius red staining was visualized under polarized light microscopy.

Cardiac histology staining was analyzed with a semiautomated image processing pipeline as previously described[68]. Briefly, the pipeline was based on ventricular thickness and HSL color differentiation. HLS color (yellow: H30°-90°, $S = 0.1–1.0$, $L = 0.1–0.93$) and (red: H330°-30°, $S = 0.1–1.0$, $L = 0.1–0.93$) were used to identified health myocardium and collagen regions respectively. The border zone was defined as the collagen fraction alteration ranging between tissues comprised of 85% collagen to 15% collagen.

## WGA staining

Wheat germ agglutinin staining was performed with WGA (Thermo-Fisher) per the manufacturer's protocol. Briefly, tissue sections were washed to move OCT, fixed with 4% paraformaldehyde, and then stained in 50 μg/mL WGA for 30 min at room temperature. Short-axis imaging of cardiomyocytes was taken in the uninjured myocardium. The Cardiomyocyte area was calculated using ImageJ software (ImageJ 1.52 v).

## Proteomics and data processing

Proteomics was performed by using the label-free proteomic method[68]. Tissues were lysed with RIPA (Beyotime) plus proteinase and phosphatase inhibitors (Beyotime) on ice. After precipitation with acetone at -20 °C overnight, proteins were reduced through DTT (ThermoFisher, >99%) for 1 h at 55 °C and incubated with IAA (ThermoFisher) for alkylation at room temperature in the dark for 30 min. Afterward, a Trypsin Protease (ThermoFisher) was used to digest proteins overnight at 37 °C, followed by an FA (ThermoFisher, >99%) to terminate the digestion. The extracted peptides were desalted using MacroSpin C18 column (Nest Group), lyophilized via vacuum concentrator, dissolved with 0.1% FA, and quantified equally using ThermoFisher quantitative fluorometric peptide assay before performing LC-MS/MS analysis. For data-independence-analysis (DIA), a mixture of peptides obtained from each sample was processed according to the High-pH reversed-phase fractionation spin column (ThermoFisher) manufacturer's instruction. Q-Exactive™ Plus Hybrid Quadrupole-Orbitrap™ (ThermoFisher) was used to identify the peptides. The MS raw files were analyzed with Skyline. The different expression proteins were defined with |fold-change| >1.5 and $p < 0.05$ via the Limma package. Gene ontology (GO) terms were analyzed via DAVID (https://david.ncifcrf.gov/), and KEGG was performed on the KEGG mapper (https://www.kegg.jp/). Gene Set Enrichment Analysis was processed on GSEA (http://www.gsea-msigdb.org/). Protein and protein interaction analysis was processed using Cytoscape (https://cytoscape.org/). IPA upstream regulator analysis was conducted via Ingenuity Pathway Analysis software.

## Apoptosis assays

For AnnexinV/PI staining, a commercial kit (BD Bioscience) was used. Cells with OGD/R disposal were digested into single-cell suspension. The staining solution was added into the suspension and then incubated for 20 min at room temperature in the dark. Samples were assessed on a flow cytometer and data was analyzed using Flow-Jov_10.7.1. The gating strategy was presented in Supplementary Fig. 37b. For Tunnel staining, cells were fixed with 4% paraformaldehyde and permeabilized with 0.5% Triton X-100 PBS solution, stained using a TUNEL kit (Beyotime), and visualized via a fluorescence microscope.

## Immunofluorescent staining

For cell samples, cells were fixed with 4% paraformaldehyde and permeabilized with 0.5% Triton X-100 PBS solution. After blocking with 5% bovine serum albumin (BSA, Beyotime, >99%), cells were incubated overnight at 4 °C with indicated primary antibodies (γ-H2AX, 1:200, Abcam; cTnT, 1:500, Abcam; β-actin, 1:1000, Proteintech; Alexa Fluor™ Plus 647 Phalloidin, 1:1000, ThermoFisher). Incubation with fluorescently coupled secondary antibodies (Goat anti-Rabbit IgG (H + L) Cross-Adsorbed Secondary Antibody Alexa Fluor™ 594, 1:1000, ThermoFisher; Goat anti-Mouse IgG (H + L) Cross-Adsorbed Secondary Antibody Alexa Fluor™ 488, 1:1000, ThermoFisher; Goat anti-Mouse IgG (H + L) Cross-Adsorbed Secondary Antibody Alexa Fluor™ 594, 1:1000, ThermoFisher) was followed by further washing in PBS. After washing, nuclei were visualized with 4,6-diamidino-2-phenylindole (DAPI, ThermoFisher). For histological samples, 7 μm tissue cryosections were washed with PBS to remove OCT. After fixation and permeabilization blocking, tissues were stained with the indicated primary antibody overnight. Afterward, the tissue was rinsed, stained with fluorescently coupled secondary antibodies for 1 h at room temperature, and then mounted with DAPI. The slides were visualized using a fluorescence microscope.

## Western blot

Proteins were extracted from in vitro cultured cells and mouse heart tissue with RIPA (Beyotime) plus proteinase and phosphatase inhibitors (Beyotime). BCA Protein Assay Kit (ThermoFisher) was used to measure protein concentration. After separation by SDS–PAGE, proteins were transferred to Polyvinylidene fluoride (PVDF) membranes. Membranes were then blocked with 5% skim milk in Tris-buffered saline/Tween20 (TBST, pH 7.4–7.6) and incubated with appropriate primary antibodies at 4 °C overnight. After being washed, the

membranes were incubated with HRP-conjugated IgG antibody (Anti-Rabbit IgG(H + L) antibody, 1:10000, Proteintech; Anti-Mouse IgG(H + L) antibody, 1:10000, Proteintech) for 2 h at room temperature. The membranes were blotted with SuperSignal West Pico PLUS (Thermo-Fisher). The original band images of western blot were presented in Supplementary Fig. 38. The following primary antibodies were used: β-actin (1:10000, Proteintech), Hsp90 (1:5000, Proteintech), Bax (1:1000, Abcam), C-casp3 (1:1000, Proteintech), Bcl-2 (1:2000, Proteintech), Jnk (1:2000, Proteintech), p-Jnk (1:1000, Abcam).

## Quantitative real-time PCR

Total RNA was isolated from cultured cells and heart tissue with Total RNA Isolation Kit (Vazyme) following the manufacturer's protocol. First-strand cDNA was synthesized with a cDNA Synthesis Kit (Vazyme). Quantitative real-time PCR was performed using qPCR Master Mix (Vazyme). The targeted gene expression levels were normalized to 18s rRNA level. The following primers were used: Mouse-*Ho-1*: 5′- CTCTC TTCTCTTGGGCCTCTA -3′, 5′- TGTCAGGTATCTCCCTCCATTC -3′; Mouse-*Nox2*: 5′- GAAAACTCCTTGGGTCAGCACT -3′, 5′- ATTTCGACA-CACTGGCAGCA -3′; Rat-*Ho-1*: 5′- TTTCAGAAGGGTCAGGTGTCC -3′, 5′-CTGCTTGTTTCGCTCTATCTCC -3′; Rat-*Nox2*: 5′- CCTGTATGTGGCTG TGACTC -3′, 5′- TCAAAGTAAGACCTCCGAATGG -3′; *18s*: 5′- CCTGTATG TGGCTGTGACTC -3′, 5′- TCAAAGTAAGACCTCCGAATGG -3′.

## ICP-MS detection

Collected tissues were quantified, mixed with aqua regia, and incubated at 80 °C until completely dissolved. The solution was centrifuged at 12,000 × g for 20 min and then collected supernatant. The supernatant was detected by an inductively coupled plasma-mass spectrometer (Agilent 7900 spectrometer).

## Statistical analysis

All the data were presented as mean ± SEM. The student *t*-test was performed for comparison within 2 groups, and One-way ANOVA or Two-way ANOVA followed by a post hoc Bonferroni test was used among multiple groups comparisons. The significance level was set at $p < 0.05$. Calculations were performed using SPSS software 19.0 (SPSS Inc., USA) and GraphPad Prism 9.0 (GraphPad Prism Software, Inc., San Diego, CA).

## Reporting summary

Further information on research design is available in the Nature Portfolio Reporting Summary linked to this article.

## Data availability

The mass spectrometry proteomics data generated in this study have been deposited in the ProteomeXchange Consortium under accession code PXD048584. All data needed to support the conclusions in the paper are available within the article and the Supplementary Information Files. Any other data related to this work are available from the corresponding author upon request. Source data are provided as a Source Data file in this paper.

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

## Acknowledgements

We gratefully thank Prof. Congfeng Xu and Dongmei Shi (Department of Cardiology, Shanghai Sixth People's Hospital Affiliated to Shanghai Jiao Tong University School of Medicine) for their assistance. This study is supported by grants from the Double-Employment Principal Investigator Project of Shanghai Sixth People's Hospital Affiliated to Shanghai Jiao Tong University School of Medicine (No.X01225 to C.S.), the National Natural Science Foundation of China (No.8227050754 to X.J.; No.22077079 to X.D.; No.821715479 to C.S.). The Figs. 6a, 7a, 8a, and i in this paper were created with BioRender.com and/or by adapting from BioRender.com templates (Publication License # VR26CDXMOB).

## Author contributions

T.Y., C.C., D.W., C.S., X.J., X.W. and X.D. designed research; T.Y., C.C., D.W., C.H., Z.Y., X.W. and Y.C. performed research; T.Y., C.C., D.W., X.W. analyzed data; T.Y., C.C., D.W., X.W., Z.Y. draw the illustrative figures; T.Y., C.C., D.W. wrote the paper.

## Competing interests

The authors declare no competing interests.
