## [Peer Review File · Nature Communications]

Reviewers' Comments:

Reviewer #1:

Remarks to the Author:

In this paper, the authors prepared a Pt single-atom nanoenzyme characterized with a structure of Pt-N-C (PtsaN-C) to explore their effects on myocardial ischemia reperfusion injury. It revealed that PtsaN-C possessed excellent ROS elimination activity under myocardial ischemia reperfusion injury by mimicking multienzyme activities including SOD-like, CAT-like, POD-like, and GPx-like properties. Overall, PtsaN-C exerted excellent cardioprotective effects, which can decrease infarct volume and improve cardiac function by ameliorating cell apoptosis. This article is interesting and has certain merits. However, there are still a lot of deficiencies and errors in the process of experimental design and article writing. The significance of this paper is also not expounded sufficiently. The authors need to highlight this paper's innovative contributions. It will require significant revisions to make the paper suitable for publication.

- (1) In the section of introduction, the antioxidant clinical therapy for myocardial ischemia reperfusion injury is not clearly described. It is suggested that the author summarize the current clinical application status of antioxidants to point out the advantages and disadvantages of antioxidants.
- (2) Among numerous single-atom nanozymes (SAM), why did the authors choose PtsaN-C to study its therapeutic effect on myocardial ischemia reperfusion injury? The significance and innovation of this paper is not expounded sufficiently.
- (3) In the section of introduction, When the author compares the structure of PtsaN-C and PtnpN-C, please add the corresponding diagram. And explain why the Pt-N₄ structure has a more efficient catalytic and antioxidant activity.
- (4) For Multienzyme-mimicking property of PtsaN-C, please explain what structure of PtsaN-C makes it has SOD-like, CAT-like, POD-like, and GPx-like activities.
- (5) PtsaN-C possessed superior cytocompatibility in H9C2 cells compared to bare Pt nanoparticles. Please add the cytotoxicity experiment data of bare Pt nanoparticles.
- (6) For figure 5i, the field of view of the TUNEL staining images intercepted by the author are too small. The results are not convincing.
- (7) In vivo, PtsaN-C was injected in situ to treat myocardial ischemia reperfusion injury. Why not choose intravenous injection? How about the targeting ability of PtsaN-C to infarcted myocardium?
- (8) The image of figure 6j are blurred, and please replace it.
- (9) Please describe the result of Figure 6m in detail.
- (10) For figure S27, the WB results showed that OGD/R induced an increase in p-JNK expression, while JNK expression did not change. But the immunofluorescent cell staining results showed that OGD/R induced an increase in JNK expression. Is this contradictory?
- (11) For figure S27, the WB results need housekeeping protein.
- (12) For figure 8d, PtsaN-C was also accumulated in lung, and please explain this result. In addition, is PtsaN-C distributed in the spleen?
- (13) In the section of discussion, it is also necessary to introduce previous research for comparison, thus highlighting the innovation of this paper. Meanwhile, please point out the shortcomings of this study and discuss the future research direction.
- (14) ROS are regarded as the essential players in myocardial ischemia reperfusion injury. Medications targeting ROS have been proven to effectively alleviate myocardial injury. Here, it is recommended to quote these two articles to add to the description about the effect of ROS on myocardial ischemia reperfusion injury. (Doi: 10.1016/j.bioactmat.2021.06.006; 10.1002/advs.202204999).

Reviewer #2:

Remarks to the Author:

In this work, the authors reported a single-atom Pt nanozyme with multienzyme-like activities, which possesses ROS-elimination performance against myocardial ischemia-reperfusion injury. Pt-N-C presented excellent biocompatibility and biosafety, making it promising for future clinical application. Similar works have been reported previously. Overall, this paper is not qualified enough to warrant publication in Nature Communications at the current stage. Some detailed

comments are listed below.

1. Many single-atom nanozymes (e.g., Rh, Co, Mn) have been reported to possess ROS-elimination performance. One question is what is the advantage of the Pt single-atom nanozyme? Compared with other nanomaterials, are there distinct advantages for restoring cellular homeostasis and preventing apoptotic progress after I/R injury?
2. The authors mentioned that the single-atom nanozyme features remarkable selectivity characteristics. How to understand the remarkable selectivity of the resultant Pt single-atom nanozyme? Please explain the reason for the multienzyme-like activities of the obtained single-atom Pt nanozyme.
3. The kinetic experiments about the GPx-like activity of nanozymes need to be given.
4. The mechanism study of nanozymes needs to be further enriched to support the conclusion, not just DFT calculations.
5. Please concise the introduction and highlight the key point of this work.
6. "Nanozyme" has been widely accepted. So, some different expressions, such as nanoenzyme should be checked.
7. The Pt loading should be characterized by ICP-MS.
8. In Figure 4C, the authors present a reaction energy diagram to compare the energy profile of H₂O₂ catalyzed by different catalysts. However, barriers typically refer to activation energies that can only be obtained by transition state calculations. The authors are recommended to perform transition state calculations for all reactions or reformulate their argument based on reaction energies instead of barriers.
9. The authors are recommended to define "int 1.....int6" in Figure 4c.

Reviewer #3:

Remarks to the Author:

Review for:

Protective Effects of Pt-N-C Nanozymes against Myocardial Ischemia² reperfusion Injury

The manuscript by Ye et al. presents the fabrication of a single atom Pt-N-C nanozymes, and their catalytic function targeting ROS, and mimicking the activity of multiple ROS reducing enzymes. Nanozymes are a class of nanomaterials that exhibit enzyme-like catalytic activity. Unlike natural enzymes, which are typically proteins, nanozymes are synthetic nanoscale structures composed of various materials, including metals, metal oxides, and carbon-based materials. Nanozymes have gained significant attention in recent years due to their unique catalytic properties and potential applications in various fields, including medicine, environmental remediation, and analytical chemistry.

The authors thoroughly characterized the PtsaN-C nanozymes to demonstrate a single atom is anchored with N-C complexes to the graphene base, and compared it with Pt nanoparticles. The structural and compositional analysis was thorough. The catalytic activity of PtsaN-C was compared with that of PtnpN-C in solution by multiple tests and it was shown to be significantly more effective in reducing OH and H₂O₂, and in mimicking the activity of several reducing enzymes (SOD, GPx, CAT, POD). Then, the authors demonstrated the utilization of the PtsaN-C nanozymes for ROS scavenging and antiapoptotic effects in vitro and in alleviating myocardial IR injury in vivo by showing improved %EF, and reduced infarct size.

It is very clear that a lot of thoughtful work was invested in this study, and its translational implications are clear. Ischemia reperfusion injury is indeed still an unsolved problem with deleterious outcomes, not only for myocardial injury, but also brain, renal, and hepatic injuries among other. Therefore, developing a working strategy that can effectively mitigate oxidative stress and scavenge ROS, would be instrumental for treatment.

However, although the results generally support the claims, there are a few comments that require further action.

Major comments

1. The authors tested ROS scavenging and anti-apoptotic effects of PtsaN-C on OGD/R-induced cell injury (Fig.5). The effects were measured 24h after reperfusion. However, it is well known that some of the processes related to ROS damage are elicited in the time scale of minutes to hours from the onset of reperfusion. Even in figure 3 it is well demonstrated that catalase and gp_x like

activities occur within minutes. Since ROS generation and oxidative stress following IRI are dynamic processes, the timing of the test should align with the hypothesis regarding when ROS levels are most critical and when the therapeutic effects of the nanoparticles are expected to be most beneficial. It seems like it would be of interest to also examine the response of the cells to the treatment in a narrower time frame.

2. In the OGD/R model, the highest PtsaN-C concentration was 10ug/mL. However, it seemed like up to 500ug/ml didn't affect cellular viability, and in the tested range the rescue effect did not saturate (referring to Fig. S16). Why didn't the authors use higher concentrations?

3. The in vitro tests were performed using H9C2 cell line, which is not highly contractile. It is expected that the generated damage in a simulated ischemia model would be more prominent than the 20% cell death shown here. Also, it is not clear from the methods section whether the media contained serum during OGD. Can the authors show indicators for contractile activity in vitro, prior and after OGD/R?

4. It is not entirely clear what was the concentration of PtsaN-C in the in vitro model. It is suggested to perform a dose response to find out the optimal concentration for rescue of the cells. What was the used concentration and how was it chosen?

5. Importantly, it is suggested that the in vitro and in vivo tests demonstrating the applicability and efficiency of PtsaN-C for alleviating ROS-induced damage, would be performed and compared with PtnpN-C.

6. Relating to the previous comment, it is important to compare the effects of the PtsaN-C particles to other nanoparticles that act as antioxidants, to highlight the novelty of this research and the relative advantage of these particles. Especially since Pt nps were recently shown to alleviate hepatic IRI as well as renal and neuronal. Therefore, it is crucial to show the advantage of the single atom particles not only in solution, but also in the in vitro and in vivo applications.

7. In Figure 6i, the tunel positive nuclei in the I/R group is about 7% while the treatment reduced it to 4%. On the other hand, the relative scar area is shown to be around 50%, and %EF as also reduced by 50%. How come the tunel positive count is so low?

8. Could the authors comment on how they selected the treatment concentration for the in vivo experiments?

9. In figure 7d, it would be beneficial to present the heat map of I/R and I/R-PtsaN-C as relative to sham, to make the comparison easier to observe by eye.

10. Although the results section is rich in information and validation of the potential therapeutic activity of the suggested nanozyme, the discussion part of the manuscript is poor. The authors didn't compare the results obtained here to previously published results (including metal nanoparticles and nanozymes) and to the history of trials and body of research where ROS scavengers / antioxidants were used in an attempt to alleviate myocardial IRI damage. Where did previous treatments fail, that this one would succeed? What is the outlook towards translating this to clinical application? Refer to the differences in the results obtained in vitro and in vivo.

11. Furthermore, there are multiple recent reviews and studies describing the use of nanozymes and single atom Pt (or other metals) as ROS scavengers, antioxidants and redox regulators. Please highlight where this study is unique and novel

Minor comments:

1. There are multiple syntax and grammar mistakes, and some text editing is needed. Pay attention to figure legends as well.

2. Regarding ROS-scavenging properties presented in figure 3, besides for panel a, it is not clear what are the used PtsaN-C concentration, and whether optimal enzyme-like activity is achieved in similar concentration for all.

3. The authors state that the activity of PtsaN-C remains stable over 40°C. However, this temperature is irrelevant for the suggested application, and it should be noted that at normal body temperature the activity is about 70% while for catalase, naturally it is above 90%. Therefore, in that sense catalase (and the other enzymes) is superior since their activity is optimal at physiological temp.

4. Legend of main Figure 6 is cut off.

5. Error bars in figure 6l showing %EF and FS are surprisingly small. This is atypical, as usually there is higher variability between animals. Were these taken from different animals, or do they represent technical replicates, of multiple measurements from the same animal? It is suggested to add individual data points.

6. Please note that acronyms are not used without prior definition. Make sure to define all

acronyms.

7. Figure 5a depicts the OGD/R procedure, depicting 6h "ischemia" and 24h reperfusion. However, in the methods section it is described as 8h with OGD and 12h reperfusion. Please make sure all information is consistent.

8. There is room for improvement in the methods section (described in the SI). There are many missing details, and it should be more accurately described, and references should be included as suitable.

Detailed Response to Reviewers

Title: “Protective Effects of Pt-N-C Nanozymes against Myocardial Ischemia-reperfusion Injury”

Authors: Tianbao Ye, Cheng Chen, Di Wang, Chengjie Huang, Zhiwen Yan, Yu Chen, Xian Jin, Xiuyuan Wang, Xianting Ding, Chengxing Shen

Corresponding authors: Xian Jin, Xiuyuan Wang, Xianting Ding, Chengxing Shen

Response to reviewers’ comments:

We thank the reviewers and editor for constructive comments which have helped us to greatly improve our research and the quality of our manuscript. Below, we carefully address the questions raised by reviewers one by one. All the changes in the revised *Manuscript* file have been marked in the **yellow highlight**.

Reviewer 1

In this paper, the authors prepared a Pt single-atom nanoenzyme characterized with a structure of Pt-N-C (PtsaN-C) to explore their effects on myocardial ischemia reperfusion injury. It revealed that PtsaN-C possessed excellent ROS elimination activity under myocardial ischemia reperfusion injury by mimicking multienzyme activities including SOD-like, CAT-like, POD-like, and GPx-like properties. Overall, PtsaN-C exerted excellent cardioprotective effects, which can decrease infarct volume and improve cardiac function by ameliorating cell apoptosis. This article is interesting and has certain merits. However, there are still a lot of deficiencies and errors in the process of experimental design and article writing. The significance of this paper is also not expounded sufficiently. The authors need to highlight this paper's innovative contributions. It will require significant revisions to make the paper suitable for publication.

Response: We would like to thank this reviewer for careful reading of our manuscript and providing constructive comments that allowed us to further improve the quality of the manuscript. We have addressed the inquiries point by point accordingly. Please find the following detailed responses to your kind comments and suggestions.

1. In the section of introduction, the antioxidant clinical therapy for myocardial ischemia reperfusion injury is not clearly described. It is suggested that the author summarize the current clinical application status of antioxidants to point out the advantages and disadvantages of antioxidants.

Response: Thanks a lot for your suggestions. According to the reviewer's suggestion, we have added current clinical application status of antioxidants in the **Introduction**, session as "... Recently, many clinical trials have demonstrated the highly beneficial effects of guanylyl cyclase enzyme activators, xanthine oxidase inhibition, and isosorbide dinitrate/hydralazine, however, there remains no effective antioxidants approved for clinical treatment of I/R injury²⁴⁻²⁶. Despite few widely used cardiovascular drugs with potential antioxidant effects, including angiotensin converting enzyme inhibitors, β -blockers and coenzyme Q10, the slow onset of action and limited treatment effectiveness hinder their widespread application as antioxidants^{25,27}. This hindrance is attributed to the poor druggability, low drug bioavailability, and non-negligible side effects^{26,28}." (Page 3, Line 23-31).

2. Among numerous single-atom nanozymes (SAN), why did the authors choose PtsaN-C to study its therapeutic effect on myocardial ischemia reperfusion injury? The significance and innovation of this paper is not expounded sufficiently.

Response: Thank you for your comment and suggestion. Since the discovery of Fe₃O₄ nanoparticles as mimics of peroxidase enzymes, various nanomaterials capable of emulating the functions of natural enzymes (i.e., nanozymes) have been explored in the fields of chemistry, biology, and medicine. Single-atom nanozymes (SANs), providing nearly 100% atom dispersion and maximized metal utilization, inherit advantages from homogeneous catalysts (isolated catalytically active sites), heterogeneous catalysts (outstanding stability and recycling use) and natural enzymes (superior activity and selectivity) [*Chem. Rev.* 2019, 119, 1806–1854; *Chem Soc Rev.* 2022 May 10;51(9):3688-3734]. In comparison to the traditional nanozyme, SANs have unique electronic/geometrical structural advantages when used for bio-application [*Adv Mater.* 2020 Feb;32(8):e1905994; *Adv Mater.* 2023 Feb 11:e2211724; *Adv. Funct. Mater.* 2020, 30, 1905410]: (I) The fully exposed unsaturated coordination active atoms combined with the charge-transfer effect induced or tailored by strong metal-support interaction can significantly aggrandize the intrinsic activity of this active site and amplify the signal with high sensitivity, meeting the

requirement of the efficient catalysis; (II) Like homogeneous enzyme, the well-defined homogenized active sites with simple coordination of SANs make it clear for the accurate identification/ engineering the active sites and demonstration the catalytic mechanism, providing an ideal model to reveal the structure-activity relationship and mimic the structure of natural enzymes; (III) SANs, especially carbon-supported catalysts (e.g., M–N_x–C, M = Fe, Cu, Zn, and others) with similar active MN_x sites of natural metalloenzymes, hold great potential for mimicking the structures and catalytic properties of natural enzymes.

Graphene, metal and metal oxide NPs have been suggested as antioxidant enzyme-mimetic nanomaterials, scavenging ROS in biological systems. Among these, PtNPs have attracted particular interest, owing to their high efficiency and selectivity as artificial CAT, POD and SOD enzymes [*Chem Soc Rev.* 2017 Aug 14;46(16):4951-4975; *Nano Lett.* 2010 Jan;10(1):219-23]. The ability of PtNPs to quench H₂O₂ and O₂^{•-}, acting as artificial enzymes, have broadly confirmed and investigated in hepatic, renal and neuronal IR injury [*Nat Commun.* 2022 May 6;13(1):2513; *Chem Eng J.* 2021 Oct 1;421:129963; *ACS Nano.* 2019 Oct 22;13(10):11552-11560]. Additionally, PtNPs have been demonstrated to mimic the mitochondrial complex I NADH ubiquinone oxidoreductase, possessing catechol oxidase-like activity and the ability to suppress lipid peroxidation [*ACS Appl Mater Interfaces.* 2015 Sep 9;7(35):19709-17]. Besides, the antioxidant activity of PtNPs has proved to be more efficient, due to PtNPs chemical stability and resistance to aggregation even in extremely diverse and complex physiological environments [*J Nanopart Res.* 2011 Oct;13(10):5547-5555]. However, recent studies revealed that the POD and CAT mimetic activities of PtNPs are dependent on the particle diameter [*Nanoscale.* 2014 Aug 21;6(16):9618-24]. Indeed, it has been documented that by decreasing the size of catalysts, the catalytic performance in terms of its activity and selectivity will be substantially improved [*Acc Chem Res.* 2013 Aug 20;46(8):1740-8].

Based on the preference for platinum nanoparticles in the aforementioned studies and the advantages of SANs, we proposed the hypothesis that our designed Pt single-atom nanozymes with a Pt–N₄–C (Pt_{sa}N–C) structure could possess both advantages of SANs and PtNPs to exert more extraordinary antioxidative functions for ROS elimination under myocardial I/R injury by mimicking multienzyme activities including SOD-like, CAT-like, POD-like, and GPx-like properties compared with PtNPs.

According to the reviewer's comment, we have also added these additional elaborations in the revised *Manuscript* (Page 3, Line 22 - Page 4, Line 1; Page 12, Line 37 - Page 14, Line 4).

3. In the section of introduction, When the author compares the structure of PtsaN-C and PtnpN-C, please add the corresponding diagram. And explain why the Pt-N₄ structure has a more efficient catalytic and antioxidant activity.

Response: Thank you for your suggestions. According to the reviewer's suggestion, we have added the corresponding structural diagram of PtsaN-C and PtnpN-C (**Scheme 1**). The reasons why the Pt-N₄ structure has a more efficient catalytic and antioxidant activity was attributed to the small reaction energy barrier and strong electron attraction from nitrogen in Pt-N₄, which collectively enhance the ability to absorb negative electron groups (O₂^{•-}, OH⁻, etc.) for Pt-N₄. Detailed explanations are also provided in the revised section of **Result** (*In situ Raman and density functional theory calculations of CAT-like activities of PtsaN-C*)

Scheme 1. Schematic diagram of Pt single atom (PtsaN-C) and Pt nanoparticles (PtnpN-C) on graphene quantum dots.

4. For Multienzyme-mimicking property of PtsaN-C, please explain what structure of PtsaN-C makes it has SOD-like, CAT-like, POD-like, and GPx-like activities.

Response: Thank you for the reviewer's comments and suggestions. Unlike the specificity exhibited by natural enzymes, artificial nanozymes demonstrate catalytic activity in a category of analogous reactions owing to certain distinctive physicochemical properties. Regarding the active site of the PtN₄ coordination configuration nanozyme synthesized in our study, consistent confirmation of the structure was obtained through various experimental characterizations. These analyses revealed a significant charge transfer between Pt and the coordinating N in the structure,

resulting in a distinctive electron-deficient state at the Pt site (**Supplementary Fig. 15**). Combining DFT theoretical calculations, it was demonstrated that this active site is favourable for the adsorption of reactive oxygen species (ROS) such as $O_2^{\bullet-}$, OH^- or H_2O_2 . The adsorption energies for $O_2^{\bullet-}$ and H_2O_2 were found to be -0.75 eV and -0.96 eV, respectively (**Figure R1a**). As illustrated in **Figure R1b**, the reaction pathways of superoxide dismutase (SOD), catalase (CAT), peroxidase (POD), and glutathione peroxidase (GPx), despite the participation of H^+ , TMB, or GSH, predominantly involve the transformation of ROS moieties, primarily $O_2^{\bullet-}$ and H_2O_2 . For degradation pathways of ROS with similar reactants, PtsaN-C can play a comparable role within a biological context.

Figure R1. a Schematic diagram of adsorption energy and electronic interaction between $O_2^{\bullet-}$ and H_2O_2 in PtN_4 coordination from PtsaN-C. **b** The reaction pathways of superoxide dismutase (SOD), catalase (CAT), peroxidase (POD), and glutathione peroxidase (GPx).

5. PtsaN-C possessed superior cytocompatibility in H9C2 cells compared to bare Pt nanoparticles. Please add the cytotoxicity experiment data of bare Pt nanoparticles.

Response: Thank you very much for the kind suggestion. Per your suggestion, we have further conducted an additional cytotoxicity experiment of bare Pt nanoparticles, and our in vitro data indicated that the cell viability obviously decreased when the concentration of bare Pt nanoparticles reached $50 \mu\text{g/ml}$, and the higher concentration of bare Pt nanoparticles showed the more severe cellular cytotoxicity (**Supplementary Fig. 16**). Previous studies have also demonstrated that the concentration of bare Pt nanoparticles more than average of $50 \mu\text{g/ml}$ might

cause vigorous cellular injuries [*Neurochem Res.* 2021 Dec;46(12):3325-3341; *Acta Biomater.* 2022 Oct 15;152:546-561]. As the concentration of PtsaN-C required to induce cytotoxicity was far higher than bare Pt nanoparticles, PtsaN-C thus has superior cytocompatibility than bare Pt nanoparticles. According to the reviewer's comment, we have also added the additional cytotoxicity experiment data of bare Pt nanoparticles into the revised *Supplementary Information* (Page 16, Line 1-6).

Supplementary Figure 16. Cell viability was obtained at different concentrations of PtsaN-C (**left**) and bare Pt nanoparticles (**right**) in H9C2 cells using CCK-8 kit (n=6 for each group). Data were analysed with One-way ANOVA with Bonferroni post hoc test, and represented Mean±SEM. The difference was in comparison to 0 µg/ml group, ***p<0.001. n.s., no significance.

6. For figure 5i, the field of view of the TUNEL staining images intercepted by the author are too small. The results are not convincing.

Response: Thank you for your detailed reading and comments. We have therefore updated the TUNEL staining images with a larger field of view in **Figure 5i** in the revised *Manuscript*.

Figure 5. Apoptosis of cells accessed by TUNEL assay. The white arrows denote the apoptotic cells.

7. In vivo, PtsaN-C was injected in situ to treat myocardial ischemia reperfusion injury. Why not choose intravenous injection? How about the targeting ability of PtsaN-C to infarcted myocardium?

Response: Thanks for the valuable discussions and inquiries. Generally speaking, biomaterials must pass through the lungs into the systemic circulation before accessing the heart via intravenous administration. Nanoparticles make use of passive or active targets for reaching infarcted heart tissue. Passive targeting is based on the heart's enhanced permeability and retention effect (EPR) after ischemia reperfusion injury [*Nano Lett.* 2011 Feb 9;11(2):694-700], whereas in active targeting strategies, nanoparticles are functionalized with infarction targeting moieties such as cluster of differentiation 34 (CD34) or the angiotensin II receptor (AT1), which are expressed by cardiomyocytes [*Nat Nanotechnol.* 2017 Sep 6;12(9):845-855]. While passively targeting biomaterials leverage the EPR effect for infarct localization, they are highly subject to higher off-target effects and nonspecific binding, lack long-term retention, and need to be delivered within

the first few days of reperfusion before the leaky vasculature phenotype is significantly attenuated [Adv Mater. 2023 Mar 29:e2300603]. Therefore, biomaterials dependent on passive targeting must be administered at high doses in order to achieve therapeutic effects, which could potentially increase the occurrence of side effects as well as thrombus and embolization.

PtsaN-C, without active targeting modification, relies on passive diffusion for reaching the infarcted heart tissue through intravenous injection. Our results showed that PtsaN-C was accumulated largely in liver and kidney, with only 10.2% ID g⁻¹ enrichment in the injured heart via intravenous injection (**Figure R2**). This implied bare PtsaN-C had a noticeable off-target effects after intravenous administration. Therefore, intramyocardial injection lowers probability of off-target effects, reducing the risk of side effects. Besides, cardiac in situ administration can be done with catheter during PCI surgery in clinic.

Figure R2. Biodistribution of PtsaN-C detected by ICP-MS. Biodistribution of PtsaN-C was measured at 6 hours after I/R injury. PtsaN-C was administrated with intravenous injection at the time of reperfusion. Data represent Mean±SEM, n=3 for each group.

8. The image of figure 6j are blurred, and please replace it.

Response: Thank you for your kind comments. We apologize for the unclear graphic display. We have replaced the TUNEL staining images for better visibility (**Figure 6j**). In addition, cardiomyocytes were identified by staining for cTnT (green). We also provided more clear images in the **Figure R3**.

Figure 6. Representative image of apoptotic cells evaluated by a TUNEL assay (n=4-6 for each group)

Figure R3. Representative image of apoptotic cells evaluated by TUNEL assay.

9. Please describe the result of Figure 6m in detail.

Response: Thank you for your suggestion. Myocardial I/R injury causes cardiomyocytes apoptosis, inflammatory response, and scar formation, ultimately leading to adverse left ventricular remodeling. To quantify infarct morphology (infarct size, border zone, and scar

thickness) precisely from histological stained sections, a semi-automated image processing pipeline was built as previously described (**Figure R4**) [*JCI Insight*. 2019 Sep 19;4(18):e131545; *Circulation*. 2021 Mar 30;143(13):1317-1330]. Briefly, the cross section of the heart is detected with inner (ventricular lumen) and outer (myocardial wall) boundaries. Local ventricular wall thickness was computed using a Eulerian solution to a pair of linear partial differential equations over the histological domain. Then wall thickness is calculated by solving the Laplace equation for a given harmonic function f between the inner and outer boundaries [*IEEE Trans Med Imaging*. 2003 Oct;22(10):1332-9]. Subsequently, the histological sections are subdivided into 120 circumferential partitions. The area fractions of myocardium (yellow: $H30^\circ-90^\circ$, $S = 0.1-1.0$, $L = 0.1-0.93$) and collagen (red: $H330^\circ-30^\circ$, $S = 0.1-1.0$, $L = 0.1-0.93$) within each partition is identified using colorimetric segmentation in a HSL color space. Based on the identified area fractions, border zone is characterized with the red/yellow ratio ranging from 0.15 to 0.85; infarct zone is defined as red/yellow ratio more than 0.85; while the remote zone is identified as red/yellow ratio less than 0.15. Finally, the ratio of infarct length to total circumferential length (in degrees) was used to estimate the percent circumference of the infarct region.

Figure R4. schematic illustration of semi-automatic histological image analysis pipeline. Picrosirius red stained short-axis sections of infarcted hearts are analysed to quantify infarct morphology. The thickness between the identified inner and outer boundaries is computed. A total of 120 circumferential partitions are defined, and colorimetric segmentation of yellow and red pixels is performed. Infarct borders are then defined based on intersections between red/yellow

area fractions; infarct and non-infarct length and thickness are calculated following semi-automatic identification.

In current study, administration of PtsaN-C significantly reduced the infarct size using the custom-built image processing pipeline according to sirius red staining at day 14 post injury (**Figure 6m, Supplementary Fig. 27b**). Moreover, border zone, as a most vulnerable area of heart after MI, bears gradual elevation of wall stress and occurs scar expansion, which in turn damages uninjured cardiomyocytes in remote area [*Circulation. 2021 Mar 30;143(13):1317-1330*]. We found that administration of PtsaN-C resulted in smaller border zone expansion (**Figure 6m, Supplementary Fig. 27b**). Although no apparent difference was observed in scar thickness, treatment with PtsaN-C could hinder adverse cardiac remodeling in terms of infarct size and border transition. Accordingly, we have added these detailed elaborations in the revised *Manuscript* (*Page 10, Line 8-12*).

Figure 6. The custom-built pipeline analysis of scar thickness, infarct size, and border zone transition. Representative Masson trichrome staining (①), Picrosirius red staining (②), pseudo-colored image (③), curve for definition of corresponding regions (④), and corresponding curve of IR mice treated with PtsaN-C or not (⑤). The border zone (green dots) was defined as the transition region between collagen-dominated scar tissue (stained >85% red) and myocardium-dominated tissue (stained >85% yellow).

Supplementary Figure 27. Quantification of infarct size, and border zone transition followed by pipeline analysis (n=6 for each group). Data represent Mean±SEM and were analysed with unpaired student *t* test. ***p<0.001.

10. For Supplementary Fig. 27, the WB results showed that OGD/R induced an increase in p-JNK expression, while JNK expression did not change. But the immunofluorescent cell staining results showed that OGD/R induced an increase in JNK expression. Is this contradictory?

Response: Thank you very much for pointing this issue out. We apologize for the confusion. Jnk is activated by phosphorylated and transferred to the nucleus, initiating the transcription of the related genes. Our immunofluorescent cell staining results showed that Jnk was assembled within the nucleus after OGD/R, indicating nuclear translocation. As only p-JNK could translocate to the nuclei, this result reflected the increase in p-Jnk after OGD/R procedure. Accordingly, we have annotated that the quantification was referred to the fluorescence intensity in the nucleus in the revised *Supplementary Information (Page 25, Line 1-8)*.

Supplementary Figure 31. Results of immunofluorescent cell staining analysis of Jnk (left), and relative quantification of fluorescent intensity in nuclei (right). Data represent Mean±SEM and were analysed with One-way ANOVA with Bonferroni post hoc test. * represent the comparison to Unstim group, *** $p < 0.001$. \$ represent the difference between I/R and I/R+PtsaN-C groups, \$\$\$ $p < 0.001$.

11. For Supplementary Fig. 27, the WB results need housekeeping protein.

Response: Thanks for the comment. According to the reviewer's suggestion, we have added *heat shock protein 90* (HSP90) as an internal reference protein in **Supplementary Fig. 27a** in the revised *Supplementary Information* (Page 25, Line 1).

Supplementary Figure 27. Representative western blot of p-Jnk proteins in cells with different dispositions (left) and corresponding quantification (right). Data represent Mean±SEM and were analysed with One-way ANOVA with Bonferroni post hoc test. * represent the comparison to Unstim group, *** $p < 0.001$. \$ represent the difference between I/R and I/R+PtsaN-C groups, \$\$\$ $p < 0.001$.

12. For figure 8d, PtsaN-C was also accumulated in lung, and please explain this result. In addition, is PtsaN-C distributed in the spleen?

Response: Thank you for your comment. According to our ICP-MS analysis, there was very few enrichments of PtsaN-C in lung. However, after comprehensive consideration lung weight and administration dosage of PtsaN-C, the normalized dosage distribution of PtsaN-C in lung was about 1.7 % ID g⁻¹, indicating PtsaN-C accumulated in lung with very small amounts. The accumulation of PtsaN-C in the lung might be attributed to the following reasons: (1) mechanical lung capillary retention as PtsaN-C could pass through the lungs upon blood circulation [*Drug Deliv.* 2017 Nov;24(1):1124-1138], (2) circulating free PtsaN-C might be phagocytised by tissue resident macrophages in lung, (3) free PtsaN-C remaining due to insufficient tissue perfusion during sample collection.

$$\text{percentage of injected dose per gram of tissue (\% ID g}^{-1}\text{)} = (\text{Measured accumulated Pt per gram tissue} / \text{Injected Pt}) \times 100.$$

Per your kind suggestion, we have further investigated the distribution of PtsaN-C in spleen. Our results showed the accumulation of PtsaN-C was negligibly low, and gradually decreased back to normal over time with a peak on the first day after injection (about 1.9% ID g⁻¹) (**Supplementary Fig. 35a**). The trace accumulation of PtsaN-C in the spleen might be due to opsonization and phagocytose of monocytes and macrophages, as well as reasons described above. Besides, no necrosis, inflammatory infiltration, hemorrhage, or fibrosis were found in the spleen after administration of PtsaN-C (**Supplementary Fig. 35b**). These results demonstrated that despite very few accumulations in spleen, PtsaN-C exhibited insignificant in vivo toxicity. These additional results have also been added in the *Manuscript* (*Page 12, Line 10-12*) and *Supplementary Information* (*Page 27, Line 1-4*).

Supplementary Figure 35. a ICP-MS analysis of spleen at indicated time points. **b** In vivo evaluation of the toxicity of PtsaN-C to spleen by histological analysis of H&E and Masson trichrome staining at day 14 post administration. Data represent Mean±SEM.

13. In the section of discussion, it is also necessary to introduce previous research for comparison, thus highlighting the innovation of this paper. Meanwhile, please point out the shortcomings of this study and discuss the future research direction.

Response: We appreciate very much for your professional suggestion. Accordingly, we have adjusted the section of discussion, according to reviewer’s reminding. (*Page 12, Line 37- Page 14, Line 11*)

14. ROS are regarded as the essential players in myocardial ischemia reperfusion injury. Medications targeting ROS have been proven to effectively alleviate myocardial injury. Here, it is recommended to quote these two articles to add to the description about the effect of ROS on myocardial ischemia reperfusion injury. (Doi: 10.1016/j.bioactmat.2021.06.006; 10.1002/advs.202204999).

Response: Thank you very much for the kind suggestion. According to the reviewer’s constructive suggestion, we have cited these important references in the revised *Manuscript* (*Reference 26, 28*)

Reviewer 2

In this work, the authors reported a single-atom Pt nanozyme with multienzyme-like activities, which possesses ROS-elimination performance against myocardial ischemia-reperfusion injury. Pt-N-C presented excellent biocompatibility and biosafety, making it promising for future clinical application. Similar works have been reported previously. Overall, this paper is not qualified enough to warrant publication in Nature Communications at the current stage. Some detailed comments are listed below.

Response: We would like to express our gratitude for the reviewer's time in reading and providing constructive suggestions as well as critical comments, which are highly helpful and important for further improving the manuscript quality. We have performed more systematic experiments to highlight the advantage and mechanism of PtsaN-C in ROS elimination in this study. Please find the following detailed responses to your kind comments and suggestions.

1. Many single-atom nanozymes (e.g., Rh, Co, Mn) have been reported to possess ROS-elimination performance. One question is what is the advantage of the Pt single-atom nanozyme? Compared with other nanomaterials, are there distinct advantages for restoring cellular homeostasis and preventing apoptotic progress after I/R injury?

Response: Thank you for your comments. It has been a fact that Pt nanoparticles (PtNPs) have attracted particular interest, owing to their high efficiency and selectivity as artificial CAT, POD and SOD enzymes [*Chem Soc Rev.* 2017 Aug 14;46(16):4951-4975; *Nano Lett.* 2010 Jan;10(1):219-23; *Nanoscale*, 2016, 8, 3739–3752]. The ability of PtNPs to quench H_2O_2 and $\text{O}_2^{\bullet-}$, acting as artificial enzymes, have broadly confirmed and investigated in hepatic, renal and neuronal IR injury [*Nat Commun.* 2022 May 6;13(1):2513; *Chem Eng J.* 2021 Oct 1;421:129963; *ACS Nano.* 2019 Oct 22;13(10):11552-11560]. Additionally, PtNPs have been demonstrated to mimic the mitochondrial complex I NADH ubiquinone oxidoreductase, possessing catechol oxidase-like activity and the ability to suppress lipid peroxidation [*ACS Appl Mater Interfaces.* 2015 Sep 9;7(35):19709-17]. Besides, the antioxidant activity of PtNPs has proved to be more efficient, due to PtNPs chemical stability and resistance to aggregation even in extremely diverse and complex physiological environments [*J Nanopart Res.* 2011 Oct;13(10):5547-5555]. PtNPs also exhibit distinct stability at various temperatures and *pH*. However, recent studies revealed that the multienzymes mimetic activities of PtNPs are dependent on the particle diameter [*Nanoscale.*

2014 Aug 21;6(16):9618-24]. Indeed, it has been documented that by decreasing the size of catalysts, the catalytic performance in terms of its activity and selectivity will be substantially improved [Acc Chem Res. 2013 Aug 20;46(8):1740-8]. Since single-atom nanozymes, providing nearly 100% atom dispersion and maximized metal utilization, inherit advantages from homogeneous catalysts (isolated catalytically active sites), heterogeneous catalysts (outstanding stability and recycling use) and natural enzymes (superior activity and selectivity) [Chem. Rev. 2019, 119, 1806–1854; Chem Soc Rev. 2022 May 10;51(9):3688-3734]. They have unique electronic/geometrical structural advantages when used for bio-application [Adv Mater. 2020 Feb;32(8):e1905994; Adv Mater. 2023 Feb 11:e2211724; Adv. Funct. Mater. 2020, 30, 1905410]. Among all single-atom systems, Pt single-atom catalysts have been demonstrated to exhibit significantly boosted catalytic activity over that of Pt/C catalysts and ultrasmall Pt clusters [Nat. Chem. 2011, 3, 634; ACS Nano. 2019 Oct 22;13(10):11552-11560]. Given that, we proposed the hypothesis that designed Pt single-atom nanozymes with a Pt-N₄-C (PtsaN-C) structure could show impressive overall catalytic functions to the greatest extent. **Indeed, PtsaN-C not only inherited the strengths of PtNPs with respects to multienzymes mimetic activities and high stability, but exerted the advantage of single-atom catalysts with boosted catalytic efficiency and natural enzymes-like selectivity (Figure 3). Further, systematically compared the PtsaN-C with recently reported metal-based nanozymes possessing ROS-elimination performance, PtsaN-C exhibited a most catalytic efficiency in terms of the Km and turnover number (TON) (Figure 3i and Supplementary Table 6). Additionally, PtsaN-C presented good biosafety and excellent biocompatibility of PtsaN-C.** Thus, PtsaN-C had strengths for use in more complex organisms.

Our in vivo results have verified that PtsaN-C possessed dramatic therapeutic effects on antioxidant and anti-apoptosis in mice with myocardial I/R injury. PtsaN-C could scavenge burst ROS mediated by I/R injury efficiently to restore cellular homeostasis and preventing apoptosis. However, in terms of in vivo validation, it is not reasonable to directly compare our experimental results with the recently reported nanoenzymes activity due to differences in reagents, methodology, animal batches, and other factors. However, as natural catalase can effectively scavenge free radical and H₂O₂ in the organism, we used catalase (as a positive control) to compare the therapeutic effects with PtsaN-C. As shown in **Supplementary Fig. 23** and **Figure 6h**, our results demonstrated that PtsaN-C presented greater therapeutic advantage than that of catalase, in

terms of reducing cardiac oxidative stress, myocardium damage and inflammation level. Taken together with the catalytic performance and therapeutic efficacy, we believe that PtsaN-C possess distinct advantages for maintaining cellular homeostasis and alleviating apoptosis during myocardial I/R injury compared with other nanomaterials.

We have illustrated the advantage of PtsaN-C in the revised *Manuscript* (*Page 3, Line 22 - Page 4, Line 1; Page 12, Line 37 - Page 13, Line 23*).

2. The authors mentioned that the single-atom nanozyme features remarkable selectivity characteristics. How to understand the remarkable selectivity of the resultant Pt single-atom nanozyme? Please explain the reason for the multienzyme-like activities of the obtained single-atom Pt nanozyme.

Response: Thank you very much for your comment. Among single metal atom nanozymes (SANs), the atomic transition metal-nitrogen coordination centers (M-N_x, M = Fe, Cu, Zn, and others) anchored on supports are chemically, geometrically and electronically similar to the metal-catalyzed active sites of natural enzymes [*Adv. Funct. Mater.*, 2020, 30, 1905410; *J. Am. Chem. Soc.*, 2019, 141, 12005–12010]. In this regard, SANs hold great potential for performing ideal catalytic activity and selectivity to functionally mimic and even outperform the natural enzymes [*Chem Soc Rev.* 2022 May 10;51(9):3688-3734].

Unlike the specificity exhibited by natural enzymes, artificial nanozymes demonstrate catalytic activity in a category of analogous reactions owing to certain distinctive physicochemical properties. Regarding the active site of the PtN₄ coordination configuration nanozyme synthesized in our study, consistent confirmation of the structure was obtained through various experimental characterizations. These analyses revealed a significant charge transfer between Pt and the coordinating N in the structure, resulting in a distinctive electron-deficient state at the Pt site (**Supplementary Fig. 15**). Combining DFT theoretical calculations, it was demonstrated that this active site is favorable for the adsorption of reactive oxygen species (ROS) such as O₂^{•-}, OH- or H₂O₂. The adsorption energies for O₂^{•-} and H₂O₂ were found to be -0.75 eV and -0.96 eV, respectively (**Figure R1a**). As illustrated in **Figure R1b**, the reaction pathways of superoxide dismutase (SOD), catalase (CAT), peroxidase (POD), and glutathione peroxidase (GPx), despite the participation of H⁺, TMB, or GSH, predominantly involve the transformation of ROS moieties,

primarily $\text{O}_2^{\bullet-}$ and H_2O_2 . For degradation pathways of ROS with similar reactants, PtsaN-C can play a comparable role within a biological context.

Figure R5. a Schematic diagram of adsorption energy and electronic interaction between $\text{O}_2^{\bullet-}$ and H_2O_2 in PtN_4 coordination from PtsaN-C. **b** The reaction pathways of superoxide dismutase (SOD), catalase (CAT), peroxidase (POD), and glutathione peroxidase (GPx).

3. The kinetic experiments about the GPx-like activity of nanozymes need to be given.

Response: Thank you for your constructive suggestion. Per your request, we have investigated the steady-state kinetic test about the GPx-like activity of PtsaN-C employing GSH as the substrates. As shown in **Supplementary Fig. 11a**, the catalytic reaction rate of PtsaN-C and PtnpN-C displayed a positive relationship with the concentration of substrates GSH within a certain range following Michaelis-Menten kinetic. The V_{max} value of PtsaN-C was $15.52 \mu\text{M} \cdot \text{min}^{-1}$, and was more than 1.5 times higher than that of PtnpN-C (**Supplementary Fig. 11b-c**). Besides, the K_m value of PtsaN-C was lower than PtnpN-C, indicating PtsaN-C had stronger the binding capacity to the substrates. These results implied PtsaN-C mimicked the GPx-like activity more efficient than PtnpN-C. Accordingly, we have provided these additional results in the revised *Manuscript* (*Page 6, Line 35-38*) and the *Supplementary Information* (*Page 14, Line 1-6*).

Supplementary Figure 11. GPx-like activity of PtsaN-C analysed by Michaelis-Menten kinetic. **a** Steady-state kinetics analysis of PtsaN-C and PtnpN-C against GSH substrate ($n=3$ for each plot). **b,c** Corresponding Lineweaver–Burk plots of GPx-like activity of PtsaN-C (**b**) and PtnpN-C (**c**) against the variation of GSH concentration ($n=3$ for each plot). Data presented as Mean \pm SEM.

4. The mechanism study of nanozymes needs to be further enriched to support the conclusion, not just DFT calculations.

Response: Thank you for your insightful suggestions. In the revised manuscript, we have incorporated additional *in situ* enhanced Raman spectroscopy results (**Figure 4a**) to provide a more comprehensive spectroscopic understanding of PtsaN-C. Phorbol myristate acetate (PMA) was utilized to facilitate ROS generation in H9C2 cell lines. Briefly, PtsaN-C and PtnpN-C were added into the culture medium respectively before PMA stimulation. Our results showed that Raman signal peaks attributed to * O ($\sim 541 \text{ cm}^{-1}$), * OOH (~ 674 and 735 cm^{-1}), and * OH ($\sim 975 \text{ cm}^{-1}$) appeared in the spectrum after PMA stimulation. In specimens with PtsaN-C, the signal peaks underwent swift degradation within a 10-minute timeframe. Conversely, in samples hosting PtnpN-C, the degradation rate was slower compared with PtsaN-C and the persistence of *O and *OOH in the milieu was still discernible even after 30 minutes. This result verified that PtsaN-C possessed stronger property of free radical scavenging than PtnpN-C (*Page 7, Line 11-21*). It is crucial to note that we do acknowledge our current comprehension of the mechanistic aspects of artificial nanozymes action lacks robust support from advanced characterization techniques. In our subsequent work, we would like to further consider integrating more mechanism studies to elucidate deeply the mechanistic details and structure-activity relationships of single-atom nanozymes.

Figure 4. *In situ* SERS spectroscopy observed H9C2 cell lines containing PtsaN-C or PtnpN-C within response time from 0 to 30 minutes after PMA stimulation. None refers to the H9C2 cell lines without nanoenzymes.

5. Please concise the introduction and highlight the key point of this work.

Response: Thank you very much for the kind suggestion, which is highly appreciated. Per your suggestion, we have concise the section of *Introduction* and highlight the significance and innovation of our work. In addition, we also introduced previous research for comparison and clinical application status of antioxidants, thus highlighting the innovation of this paper in the revised section of *Introduction* and *Discussion*.

6. “Nanozyme” has been widely accepted. So, some different expressions, such as nanoenzyme should be checked.

Response: Thank you very much for pointing this issue out. We have made corresponding modifications throughout the manuscript.

7. The Pt loading should be characterized by ICP-MS.

Response: Thank you very much for your comment. To address the reviewer’s request, we have presented the Pt loading detected by ICP-OES in previous *Supplementary Information* (**Supplementary Table 1**). As the detection limit of ICP-OES is 1ppb, this already meets the detection requirements of PtsaN-C.

8. In Figure 4C, the authors present a reaction energy diagram to compare the energy profile of H₂O₂ catalyzed by different catalysts. However, barriers typically refer to activation energies that can only be obtained by transition state calculations. The authors are recommended to perform transition state calculations for all reactions or reformulate their argument based on reaction energies instead of barriers.

Response: Thank you to the reviewer for your professional comments. We acknowledge that transition state analysis is necessary for understanding reaction kinetics of artificial nanoenzymes. We therefore have added the theoretical calculation results in the revised manuscript and conducted a new evaluation of the energy barrier of the ROS dissociation reaction pathway (**Figure 4c, Supplementary Table 7**). A detailed discussion is also provided in the revised manuscript (*Page 7, Line 27-30*) and *Supplementary Information (Page 32, Supplementary Table 7)*.

9. The authors are recommended to define “int 1.....int6” in Figure 4c.

Response: Thank you for the reviewer's kind suggestions. For ease of understanding, we have removed these codes and directly labelled the corresponding intermediate states at **Figure 4c** of the revised manuscript.

Reviewer 3

Protective Effects of Pt-N-C Nanozymes against Myocardial Ischemia² reperfusion Injury

The manuscript by Ye et al. presents the fabrication of a single atom Pt-N-C nanozymes, and their catalytic function targeting ROS, and mimicking the activity of multiple ROS reducing enzymes.

Nanozymes are a class of nanomaterials that exhibit enzyme-like catalytic activity. Unlike natural enzymes, which are typically proteins, nanozymes are synthetic nanoscale structures composed of various materials, including metals, metal oxides, and carbon-based materials. Nanozymes have gained significant attention in recent years due to their unique catalytic properties and potential applications in various fields, including medicine, environmental remediation, and analytical chemistry.

The authors thoroughly characterized the PtsaN-C nanozymes to demonstrate a single atom is anchored with N-C complexes to the graphene base, and compared it with Pt nanoparticles. The structural and compositional analysis was thorough. The catalytic activity of PtsaN-C was compared with that of PtnpN-C in solution by multiple tests and it was shown to be significantly more effective in reducing OH an H₂O₂, and in mimicking the activity of several reducing enzymes (SOD, GPx, CAT, POD). Then, the authors demonstrated the utilization of the PtsaN-C nanozymes for ROS scavenging and antiapoptotic effects in vitro and in alleviating myocardial IR injury in vivo by showing improved %EF, and reduced infarct size.

It is very clear that a lot of thoughtful work was invested in this study, and its translational implications are clear. Ischemia reperfusion injury is indeed still an unsolved problem with deleterious outcomes, not only for myocardial injury, but also brain, renal, and hepatic injuries among other. Therefore, developing a working strategy that can effectively mitigate oxidative stress and scavenge ROS, would be instrumental for treatment.

However, although the results generally support the claims, there are a few comments that require further action.

Response: We deeply appreciate your careful reading and kind consideration of our manuscript. Please find the following detailed responses to your comments and suggestions. Thanks again for your time and help.

Major comments

1. The authors tested ROS scavenging and anti-apoptotic effects of PtsaN-C on OGD/R-induced cell injury (Fig.5). The effects were measured 24h after reperfusion. However, it is well known that some of the processes related to ROS damage are elicited in the time scale of minutes to hours from the onset of reperfusion. Even in figure 3 it is well demonstrated that catalase and gpx like activities occur within minutes. Since ROS generation and oxidative stress following IRI are dynamic processes, the timing of the test should align with the hypothesis regarding when ROS levels are most critical and when the therapeutic effects of the nanoparticles are expected to be most beneficial. It seems like it would be of interest to also examine the response of the cells to the treatment in a narrower time frame.

Response: Thank you for constructive suggestion, which is highly appreciated. Previous study has demonstrated that OGD/R injury mediated generation of ROS and lipid peroxide occurred mainly hours after reperfusion in cardiomyocytes [*Circ Res. 2023 Oct 27;133(10):861-876*]. Briefly, mitochondrial ROS is remarkably elevated at 1 hour after reperfusion, and maintains higher level after 6 hours after reperfusion. Lipid peroxide obviously increases and reaches a steady stage 1 hour after reperfusion. Based on the reviewer's suggestion, we have further investigated the cellular response at 1 and 6 hours after reperfusion. First, without PtsaN-C treatment, the cellular ROS level was dramatically increased at 1 hour after reperfusion and maintained high level 6 hours after reperfusion in accordance with the literature (**Supplementary Fig. 19a**). Treatment with PtsaN-C, however, reduced the ROS to a low level within the observation time. Combined with the previous results the most obvious difference between the two groups was observed at 6 hours. Secondly, the MDA level was elevated most at few hours and peaked at 6 hours after reperfusion (**Supplementary Fig. 19b**). Administration of PtsaN-C maintained the MDA at a low level. These indicated PtsaN-C exerted ROS scavenging function beneficially during early reperfusion. Additionally, the SOD activity and GSH content, both of which assist in ROS clearance, were significantly decreased at the first few hours after reperfusion (**Supplementary Fig. 19c-d**). Instead, PtsaN-C identifiably restored the impaired SOD activity and GSH content over time. Finally, the more ROS and lipid peroxide accumulate, the more cells death occur with time [*Circ Res. 2023 Oct 27;133(10):861-876*]. The cell viability indicated the gap between the groups were most distinct at 12 hours after reperfusion over the observation time, consistent with literatures (**Supplementary Fig. 19e**). Together, these results demonstrated that the difference between groups was more apparent at the first few hours after reperfusion with

respect to ROS-indicators, while the cellular injury indicators was more distinct over time. According to the reviewer's comment, we have also provided these additional data in the revised Manuscript (Page 8, Line 22- 23) and Supplementary Information (Page 17, Line 10-18).

Supplementary Figure 19. ROS decomposition of PtsaN-C over time. **a** ROS measured by ROS Assay Kit at indicated time (n=6 for each group). **b-c** The cellular MDA (**b**), SOD (**c**), GSH (**d**) level in each group at different time after reperfusion (n=6 for each group). **e** Cell viability during OGD/R injury (n=6 for each group). Cells were treated with 10 μg/ml PtsaN-C or equivalents of vehicle. The blue bar represents the normal culture condition, and orange bar represents the timing of reperfusion after 8 hours of OGD process. The difference was in comparison to the indicated group, *p<0.05, ***p<0.001. \$ represent the comparison to normal groups, \$\$p<0.01, \$\$\$p<0.001. n.s., no significance.

2. In the OGD/R model, the highest PtsaN-C concentration was 10μg/ml. However, it seemed like up to 500μg/ml didn't affect cellular viability, and in the tested range the rescue effect did not saturate (referring to Fig. S16). Why didn't the authors use higher concentrations?

Response: Thank you for your detailed comments. To address your inquiry, we had carried out a different PtsaN-C concentration gradients to determine the optimal concentration ahead of in vitro study. To assess in which concentration PtsaN-C could exert the optimal protective potency for alleviating oxidative stress injury, cell viability was detected by CCK-8. Our results showed

that even at a concentration of 1 $\mu\text{g/ml}$, PtsaN-C could effectively suppress OGD/R induced cell injury, and the protective effects became stronger with the increase of PtsaN-C concentration. However, further increases in PtsaN-C concentration up to 50 $\mu\text{g/ml}$ did not significantly increase the protective potency compared with 10 $\mu\text{g/ml}$ (the gap between these two groups was 5.4%), which indicated 10 $\mu\text{g/ml}$ PtsaN-C could nearly quenched most intracellular ROS. Therefore, we chose the concentration of 10 $\mu\text{g/ml}$ for in vitro experiments. Accordingly, we have provided these additional data in the revised *Manuscript (Page 8, Line 16-19)* and *Supplementary Information (Page 16, Line 10- 14)*.

Supplementary Figure 17. PtsaN-C enhancing cell viability during OGD/R procedure. The cell viability was accessed after OGD/R process in H9C2 cells with different concentration of PtsaN-C (n=6 for each group). Data were analysed with One-way ANOVA with Bonferroni post hoc test, and represent Mean \pm SEM. The difference was in comparison to the indicated group, **p<0.01, ***p<0.001. n.s., no significance.

3. The in vitro tests were performed using H9C2 cell line, which is not highly contractile. It is expected that the generated damage in a simulated ischemia model would be more prominent than the 20% cell death shown here. Also, it is not clear from the methods section whether the media contained serum during OGD. Can the authors show indicators for contractile activity in vitro, prior and after OGD/R?

Response: Thank you very much for your inquiry. Compared with H9C2 cell line, cardiomyocytes demand higher energy to sustain contractile function, which relies on efficient mitochondrial oxidative metabolism for energy production instead of anaerobic glycolysis. Thus,

Mitochondria, the primary organelles for ROS generation, are more abundant in cardiomyocytes to supply energy for repeated muscle contraction [*Cell Rep.* 2015 Oct 20;13(3):533-545]. However, myocardial I/R injury can lead to mitochondrial damage, and damaged mitochondria can produce a large amount of ROS to further attack normal mitochondria, resulting in ROS burst, ultimately leading to cardiomyocyte death. As a result, the generated damage in cardiomyocytes is more prominent than that of H9C2 cells.

We apologize for the confusion in our original manuscript. According to the OGD/R procedure, cells were cultured in serum-free no glucose DMEM in an air-tight chamber with a humidified hypoxic atmosphere containing 5% CO₂ and 95% N₂ at 37 °C for 8 hours. After the deprivation, cells were then transferred to a normal incubator and cultured in serum glucose-containing DMEM for recovery 12 hours [*BMC Med.* 2019;17:42; *Nat Commun.* 2022 Nov 9;13(1):6762]. We have also added these detailed elaborations in the revised *Supplementary Information* (*Page 4, Line 25-30*).

Per your suggestion, we have also further studied the contractile activity of cardiomyocytes in vitro prior and after OGD/R. The primary cardiomyocytes were extracted following the method in *Supplementary Information* (*Page 4, Line 10-18*). Our results showed that the contractile activity of cardiomyocytes was indeed severely impaired after OGD/R (**Figure R6**). The contraction amplitude and contraction velocity of cardiomyocytes were significantly decreased post OGD/R injury, but these indicators were largely restored after treatment with PtsaN-C (**Supplementary Fig. 20a-b, Supplementary Video 1-4**). This may be attributed to the fact that PtsaN-C efficiently scavenged ROS to mitigate cell injury during OGD/R. We have also provided these additional results in the revised *Manuscript* (*Page 8, Line 28-30*) and *Supplementary Information* (*Page 18, Line 1-7*).

Figure R6. Representative images of cardiomyocyte contraction were taken using the confocal microscope.

Supplementary Figure 20. PtsaN-C restored cardiomyocyte contractile activity after OGD/R procedure. **a** Quantitative analysis of the cardiomyocyte contraction amplitude (n=3 for each group). **b** Quantification of the cardiomyocyte contraction velocity (n=3 for each group). Data were analysed with One-way ANOVA with Bonferroni post hoc test, and represent Mean±SEM. * represent the comparison to Unstim group, *p<0.05, ***p<0.001. \$ represent the difference between I/R and I/R+PtsaN-C groups, \$p<0.05, \$\$p<0.01.

4. It is not entirely clear what was the concentration of PtsaN-C in the in vitro model. It is suggested to perform a dose response to find out the optimal concentration for rescue of the cells. What was the used concentration and how was it chosen?

Response: Thank you very much for pointing this issue out. We apologize for the confusion in our original manuscript. Our previous **methods** section in the *Supplementary Information* mentioned “Following incubation for 8 hours, the media was replaced with complete growth media containing PtsaN-C (10 μ g/ml) or references, and then cells were placed in a normoxic incubator for recovery 12 hours”. Besides, we have modified schematic illustration in **Figure 5a** with PtsaN-C concentration annotated, making it easier to observe by readers. Reasons for choosing this concentration in the in vitro model were elaborately explained in the second major comments raised above.

Figure 5. Schematic diagram of experimental approach. H9C2 cells were cultured in serum-free no glucose DMEM in an air-tight chamber with a humidified hypoxic atmosphere containing 5% CO₂ and 95% N₂ at 37 °C for 8 hours. Then H9C2 cells were transferred to a normal incubator and cultured in serum and glucose-containing DMEM for recovery 12 hours. PtsaN-C was administrated to a final concentration of 10 μ g/ml during reperfusion.

5. Importantly, it is suggested that the in vitro and in vivo tests demonstrating the applicability and efficiency of PtsaN-C for alleviating ROS-induced damage, would be performed and compared with PtnpN-C.

Response: Thanks very much for your constructive suggestion. Based on the reviewer's suggestion, we have further investigated the efficiency of PtnpN-C for alleviating ROS-induced damage and compared with PtsaN-C in vitro and in vivo. To study the effects of PtnpN-C on OGD/R-induced injury model in vitro, PtnpN-C (10 μ g/ml) and PtsaN-C (10 μ g/ml) were added to complete growth media respectively after 8 hours deprivation. Although administration of PtnpN-C not only depressed cellular MDA level, but restored the SOD activity, GSH content, PtsaN-C had more distinct advantages over PtnpN-C (**Supplementary Fig. 28a-c**). Simultaneously, PtsaN-C possessed more obvious anti-apoptotic effect than PtnpN-C according to Tunel staining (**Supplementary Fig. 28d**). In vivo, PtnpN-C (10 μ l at a concentration of 0.5 μ g/ μ l) or PtsaN-C (10 μ l at a concentration of 0.5 μ g/ μ l) was given into three areas adjacent to the injury tissue with a 32-gauge needle immediately after reperfusion. Our results showed that PtsaN-C had a priority in restoring the SOD and GSH-PX activity, alleviating MDA content compared with PtnpN-C (**Supplementary Fig. 28e-g**). Myocardial zymograms, as indicators of myocardium injury, demonstrated that treatment with PtsaN-C showed obvious lower level of myocardial zymograms PtnpN-C, presenting with lower circulating level of CK, CK-MB and LDH1 compared to PtnpN-C (**Supplementary Fig. 28h-j**). Functionally, echocardiography showed the cardiac function in I/R mice treated with PtsaN-C was significantly better than that of I/R treated with PtnpN-C (**Supplementary Fig. 28k**). In summary, these in vitro and in vivo results demonstrated that PtsaN-C possessed more efficiency for alleviating ROS-induced damage compared to PtnpN-C. Accordingly, these results have also been provided in the revised Manuscript (*Page 10, Line 19-22*) and *Supplementary Information (Page 22, Line 1-10)*.

Supplementary Figure 28. PtsaN-C represents a distinct therapeutic advantage over PtnpN-C. **a-c** The levels of cellular SOD (**a**), GSH (**b**), MDA (**c**) measured after OGD/R process (n=3 for each group). **d** Apoptosis of cells detected by TUNEL assay after OGD/R process. **e-g** The levels of SOD (**e**), GSH-PX (**f**), MDA (**g**) in cardiac tissue homogenate at day 1 after I/R injury (n=4 for each group). **h-j** Serum concentrations of myocardial enzyme spectrum CK (**h**), CK-MB (**i**), and LDH1 (**j**) tested at day 1 after I/R injury (n=4 for each group). **k** Cardiac function measured by echocardiography at day 1 after I/R injury (n=4 for each group). Data represent Mean \pm SEM and were analysed with unpaired student *t* test. **p*<0.05, ***p*<0.01.

6. Relating to the previous comment, it is important to compare the effects of the PtsaN-C particles to other nanoparticles that act as antioxidants, to highlight the novelty of this research and the relative advantage of these particles. Especially since Pt nps were recently shown to alleviate

hepatic IRI as well as renal and neuronal. Therefore, it is crucial to show the advantage of the single atom particles not only in solution, but also in the in vitro and in vivo applications.

Response: Thank you for your comments and suggestions. Graphene, metal and metal oxide NPs have been suggested as antioxidant enzyme-mimetic nanomaterials, scavenging ROS in biological systems. Among these, PtNPs have attracted particular interest, owing to their high efficiency and selectivity as artificial CAT, POD and SOD enzymes [*Chem Soc Rev.* 2017 Aug 14;46(16):4951-4975; *Nano Lett.* 2010 Jan;10(1):219-23]. The ability of PtNPs to quench H₂O₂ and O₂^{•-}, acting as artificial enzymes, have broadly confirmed and investigated in hepatic, renal and neuronal IR injury [*Nat Commun.* 2022 May 6;13(1):2513; *Chem Eng J.* 2021 Oct 1;421:129963; *ACS Nano.* 2019 Oct 22;13(10):11552-11560]. Additionally, PtNPs have been demonstrated to mimic the mitochondrial complex I NADH ubiquinone oxidoreductase, possessing catechol oxidase-like activity and the ability to suppress lipid peroxidation [*ACS Appl Mater Interfaces.* 2015 Sep 9;7(35):19709-17]. Besides, the antioxidant activity of PtNPs has proved to be more efficient, due to PtNPs chemical stability and resistance to aggregation even in extremely diverse and complex physiological environments [*J Nanopart Res.* 2011 Oct;13(10):5547-5555]. PtNPs also exhibit distinct stability at various temperatures and pH. However, recent studies revealed that the POD and CAT mimetic activities of PtNPs are dependent on the particle diameter [*Nanoscale.* 2014 Aug 21;6(16):9618-24]. Indeed, it has been documented that by decreasing the size of catalysts, the catalytic performance in terms of its activity and selectivity will be substantially improved [*Acc Chem Res.* 2013 Aug 20;46(8):1740-8]. Since single-atom nanozymes, providing nearly 100% atom dispersion and maximized metal utilization, inherit advantages from homogeneous catalysts (isolated catalytically active sites), heterogeneous catalysts (outstanding stability and recycling use) and natural enzymes (superior activity and selectivity) [*Chem. Rev.* 119, 1806–1854; *Chem Soc Rev.* 51(9):3688-3734], they have unique electronic/geometrical structural advantages when used for bio-application. Among all single-atom systems, Pt single-atom catalysts have been demonstrated to exhibit significantly boosted catalytic activity over that of Pt/C catalysts and ultras-small Pt clusters [*Nat. Chem.* 3, 634; *ACS Nano.* 13(10):11552-11560]. Since single-atom nanozymes, providing nearly 100% atom dispersion and maximized metal utilization, inherit advantages from homogeneous catalysts (isolated catalytically active sites), heterogeneous catalysts (outstanding stability and recycling use) and natural enzymes (superior activity and selectivity) [*Chem. Rev.* 2019, 119, 1806–1854; *Chem*

Soc Rev. 2022 May 10;51(9):3688-3734]. They have unique electronic/geometrical structural advantages when used for bio-application [*Adv Mater.* 2020 Feb;32(8):e1905994; *Adv Mater.* 2023 Feb 11:e2211724; *Adv. Funct. Mater.* 2020, 30, 1905410]. Among all single-atom systems, Pt single-atom catalysts have been demonstrated to exhibit significantly boosted catalytic activity over that of Pt/C catalysts and ultrasmall Pt clusters [*Nat. Chem.* 2011, 3, 634; *ACS Nano.* 2019 Oct 22;13(10):11552-11560]. Given that, we proposed the hypothesis that designed Pt single-atom nanozymes with a Pt-N₄-C (PtsaN-C) structure could show impressive overall catalytic functions to the greatest extent. **Indeed, PtsaN-C not only inherited the strengths of PtNPs with respects to multienzymes mimetic activities and high stability, but exerted the advantage of single-atom catalysts with boosted catalytic efficiency and natural enzymes-like selectivity (Figure 3). Further, systematically compared the PtsaN-C with recently reported metal-based nanozymes possessing ROS-elimination performance, PtsaN-C exhibited a most catalytic efficiency in terms of the Km and turnover number (TON) (Figure 3i and Supplementary Table 6). Additionally, PtsaN-C presented good biosafety and excellent biocompatibility of PtsaN-C.** Thus, PtsaN-C had strengths for use in more complex organisms.

Our in vivo results have further verified that PtsaN-C possessed dramatic therapeutic effects on antioxidant and anti-apoptosis in mice with myocardial I/R injury. PtsaN-C could scavenge burst ROS mediated by I/R injury efficiently to restore cellular homeostasis and preventing apoptosis. However, in terms of in vivo validation, it is not reasonable to directly compare our experimental results with the recently reported nanoenzymes activity due to differences in reagents, methodology, animal batches, and other factors. Since natural catalase can effectively scavenge free radical and H₂O₂ in the organism, we used catalase (as a positive control) to compare the therapeutic effects with PtsaN-C. As shown in **Supplementary Fig. 23** and **Figure 6h**, our results demonstrated that PtsaN-C presented greater therapeutic advantage than that of catalase, in terms of reducing cardiac oxidative stress, myocardium damage and inflammation level. Taken together with the catalytic performance and therapeutic efficacy, we therefore believe that PtsaN-C possess distinct advantages for maintaining cellular homeostasis and alleviating apoptosis during myocardial I/R injury compared with other nanomaterials.

We have also illustrated the advantage of PtsaN-C in the revised *Manuscript* (*Page 3, Line 22 - Page 4, Line 1; Page 12, Line 37 - Page 13, Line 23*).

7. In Figure 6j, the tunel positive nuclei in the I/R group is about 7% while the treatment reduced it to 4%. On the other hand, the relative scar area is shown to be around 50%, and %EF as also reduced by 50%. How come the tunel positive count is so low?

Response: Thank you for your questions and discussions. The low Tunel positive nuclei in the I/R group might be attributed to the following reasons:

(1) Although cardiomyocytes occupy ~75% of the structural space of the heart, cardiomyocytes only account for approximately 25–35% of all cells in the heart [*Circ Res. 2016 Feb 5;118(3):400-9*]. Among cardiac non-myocyte cell, endothelial cells and resident mesenchymal cells (RMC) are the most abundant cell type (**Figure R7a**). As cardiomyocytes have higher oxygen consumption and poorer tolerance to hypoxia compared with non-cardiomyocytes, the death of cardiomyocytes occurs earlier and more severe than non-cardiomyocytes during I/R injury. Besides, massive inflammatory cells infiltrate into the injury area after I/R injury [*Circ Res. 2016 Jun 24;119(1):91-112*]. Therefore, the number of cardiomyocytes is small in the injured area, while non-cardiomyocytes still account for the vast majority of all cell in the heart.

(2) Cardiomyocyte death is a fundamental aspect of myocardial I/R injury. The forms of cardiomyocytes death include necrosis and regulated cell death, including necroptosis, pyroptosis, apoptosis and ferroptosis, among which necrosis is the primary mechanisms underlying cardiomyocytes death (**Figure R7b-c**) [*Nat Rev Cardiol. 2020 Dec;17(12):773-789*]. Cardiomyocyte apoptosis occurs via the intrinsic pathway, in response to DNA damage and increased ROS and cytosolic Ca²⁺ levels, or via the extrinsic pathway, in response to activation of sarcolemmal death receptors [*Am J Physiol Heart Circ Physiol. 2019 Nov 1;317(5):H891-H922*]. The apoptotic component of cell death in the myocardium is primary triggered at the time of reperfusion [*J Cell Mol Med. 2020 Apr;24(7):3795-3806*]. Besides, cardiomyocyte apoptosis is a dynamic process, and excessive and continuous inflammatory responses induced by I/R injury can further cause cardiomyocyte apoptosis over time [*Circ Res. 2016 Jun 24;119(1):91-112*]. Thus, apoptotic cardiomyocytes detected by Tunel assay was not high.

In summary, the low Tunel positive count of cardiomyocytes could be attributed to the small number of cardiomyocytes and diverse myocardial death pathways upon I/R injury.

Figure R7. a Distribution of cardiac cell types [*Circ Res.* 2016 Feb 5;118(3):400-9]. **b** Various modes of cardiomyocyte death that occur during acute myocardial ischaemia–reperfusion include necrosis and regulated modes of cell death, including apoptosis, necroptosis and pyroptosis [*Nat Rev Cardiol.* 2020 Dec;17(12):773-789]. **c** The major pathways of cell death that contribute to myocardial ischaemia and reperfusion injury. During initial oncosis, cells swell—this is reversible but can proceed to necrosis. Cardiomyocytes die primarily via a process of necrosis/necroptosis in addition to other cell-death processes such as pyroptosis [*J Cell Mol Med.* 2020 Apr;24(7):3795-3806].

In addition, to avoid interference from non-cardiomyocytes, cardiomyocytes were identified by staining for cardiac troponin T (cTnT) in the revised *Manuscript* (Figure 6j). We also provided more clearer images in the Figure R8.

Figure 6j. Representative image of apoptotic cells evaluated by a TUNEL assay (n=4-6 for each group)

Figure R8. Representative image of apoptotic cells evaluated by TUNEL assay.

8. Could the authors comment on how they selected the treatment concentration for the in vivo experiments?

Response: Thank you for your inquiry. Our in vitro experiments confirmed that PtsaN-C concentration in the range of 10-500µg/ml was a safe and effective concentration. As the mode of

administration was intramyocardial delivery, we chose the 500 μ g/ml for the upper limit to avoid possible cytotoxicity caused by high PtsaN-C concentration in suit. Hence, in our preliminary experiments, we selected 10 μ g/ml, 100 μ g/ml and 500 μ g/ml as treatment concentration. The circulating concentrations of myocardial enzyme spectrum and cardiac function indexes were used to assessed the therapeutic effects of different doses for PtsaN-C. PtsaN-C (10 μ L at corresponding concentration) was given into three areas adjacent to the injury tissue with a 32-gauge needle immediately after reperfusion [Nat Commun. 2016 Oct 27;7:13306; Circ Res. 2011 Mar 4;108(5):582-92]. Our results showed that 500 μ g/ml treatment concentration achieved the most promising therapeutic benefits (**Figure R9**). Therefore, we selected the treatment concentration 500 μ g/ml with an injected dose of 10 μ L for the in vivo experiments.

Figure R9. Therapeutic effects of different doses for PtsaN-C. **a** Serum concentrations of myocardial enzymes CK, CK-MB, and LDH1. **b** Cardiac function detected by echocardiography at day 1 post reperfusion. Data were analysed with One-way ANOVA with Bonferroni post hoc test, and represent Mean \pm SEM. The difference was in comparison to the indicated group, *p<0.05, n.s., no significance.

9. In figure 7d, it would be beneficial to present the heat map of I/R and I/R-PtsaN-C as relative to sham, to make the comparison easier to observe by eye.

Response: Thank you very much for the kind suggestion, which is highly appreciated. Based on the reviewer's suggestions, we have modified the heatmap after comparison with sham group in the revised Manuscript (**Figure 7d**).

Figure 7. Heat map illustrating the differential proteins after PtsaN-C treatment upon I/R injury. These differential proteins participate to I/R-associated pathological processes which may exert cardioprotection.

10. Although the results section is rich in information and validation of the potential therapeutic activity of the suggested nanozyme, the discussion part of the manuscript is poor. The authors didn't compare the results obtained here to previously published results (including metal nanoparticles and nanozymes) and to the history of trials and body of research where ROS scavengers / antioxidants were used in an attempt to alleviate myocardial IRI damage. Where did previous treatments fail, that this one would succeed? What is the outlook towards translating this to clinical application? Refer to the differences in the results obtained in vitro and in vivo.

Response: Thank you very much for pointing this issue out. Although nanozymes have a great prospect in effectively maintaining the redox imbalance by mimicking endogenous enzymes, and overcoming the problems of natural enzymes including low stability and poor bioavailability, research studies in the field of myocardial ischemia reperfusion treatment remain at the initial stage [Chem. Soc. Rev. 42 (14) (2013) 6060–6093]. Among the antioxidant nanozymes, metal/metal-oxide nanozymes are more commonly investigated. Thus far, cerium oxide (ceria), iron oxide, gold, and copper have been widely investigated for myocardial ischemia reperfusion injury therapy.

Although these metal/metal-oxide nanozymes can significantly reduce ROS production and protect myocardium from oxidative injury, the side effects further hinder their application [*Bioact Mater.* 2021 Jun 20;7:47-72]. Ceria nanozyme impairs the structure and function of blood vessels and exacerbate myocardial ischemia reperfusion injury due to its proinflammatory side effects during clinical application [*Nanotoxicology.* 2011 Dec;5(4):531-45]; myocardial delivery of the iron oxide nanozyme may cause iron overload which can paradoxically induce oxidative stress to exacerbate myocardial injury owing to the Fenton reaction and ferroptosis in ischemic cardiomyocytes [*Biomaterials.* 2019 Aug;211:1-13].

Serving as the substitute for natural enzymes to resist oxidative stress, Pt has shown good cytocompatibility and multienzymes-like activity including CAT-like properties, SOD-like activity, and POD-like function [*Chem Eng J.* 2021 Oct 1;421:129963; *Nat Commun.* 2022 May 6;13(1):2513]. Single-atom nanozymes (SANs), providing nearly 100% atom dispersion and maximized metal utilization, not only inherit advantages from homogeneous catalysts (isolated catalytically active sites), heterogeneous catalysts (outstanding stability and recycling use) and natural enzymes (superior activity and selectivity) [*Chem. Rev.* 2019, 119, 1806–1854; *Chem Soc Rev.* 2022 May 10;51(9):3688-3734], but also have unique electronic/geometrical structural advantages when used for bio-application. Our designed Pt single-atom nanozymes with a Pt-N₄-C (PtsaN-C) structure not only possessed multienzymes mimetic activities and high stability, but exerted the advantage of single-atom catalysts with boosted catalytic efficiency, high atomic utilization and natural enzymes-like selectivity (**Figure 3**). Besides, compared the PtsaN-C with recently reported metal/metal-oxide nanozymes attempting to alleviate ROS-mediated tissue injury, PtsaN-C exhibited a most catalytic efficiency in terms of the K_m and turnover number (TON) (**Figure 3i** and **Supplementary Table 6**). Additionally, the chemical stability of PtNPs confer the advantages in exercising more lasting antioxidant action even in extremely diverse and complex physiological environments.

Pt-N₄-C structure maximized metal utilization, indicating the metal loading was extreme low. To systematically investigated the cytotoxicity of PtsaN-C, we have further conducted an untargeted proteomics. Our results demonstrated that PtsaN-C injection alone not only caused no cytotoxicity, but could enhance the oxidoreductase activity, disulfide oxidoreductase activity, and ATPase activity in proteins, which were participated in biological processes related to cell redox homeostasis, inflammation regulation, DNA repair (**Figure 8a**, **8b** and **Supplementary Fig. 32**).

Moreover, examination of histology and serology also verified the PtsaN-C was the absence of cytotoxicity under treatment dose.

Collectively, PtsaN-C not only possesses multiple antioxidant enzyme-like properties and high efficiency of ROS clearance, but maximally reduces the biotoxicity of traditional platinum nanozymes, offering a promising therapeutic alternative for the treatment of myocardial I/R injury.

Per the reviewer's comment, we have carefully modified the discussion part in the revised Manuscript (*Page 12, Line 37 - Page 14, Line 11*).

11. Furthermore, there are multiple recent reviews and studies describing the use of nanozymes and single atom Pt (or other metals) as ROS scavengers, antioxidants and redox regulators. Please highlight where this study is unique and novel

Response: Thank you for your constructive suggestion. Per the reviewer's suggestion, we have compared the PtsaN-C with recently reported metal/metal-oxide nanozymes attempting to alleviate ROS-mediated tissue injury, PtsaN-C exhibited a most catalytic efficiency in terms of the K_m and TON (**Figure 3i** and **Supplementary Table 6**). Our study has the following uniqueness: (I) High catalytic efficiency. PtsaN-C has multienzymes activity similar to PtNPs. Besides, the fully exposed unsaturated coordination active atoms combined with the charge-transfer effect induced or tailored by strong metal-support interaction can significantly aggrandize the intrinsic activity of this active site and amplify the signal with high sensitivity, meeting the requirement of the efficient catalysis; (II) High stability. PtsaN-C not only possesses the inherent stability of Pt, but inherits advantages from heterogeneous catalysts (outstanding stability and recycling use), which makes PtsaN-C more efficient even in extremely diverse and complex physiological environments; (III) High biocompatibility. PtsaN-C, providing nearly 100% atom dispersion and maximized Pt utilization, reduces metal toxicity to the greatest extent. We have highlighted the novelty and advantage of PtsaN-C in the revised Manuscript (*Page 13, Line 24 - Page 14, Line 4*).

Minor comments:

1. There are multiple syntax and grammar mistakes, and some text editing is needed. Pay attention to figure legends as well.

Response: We apologize for the syntax and grammar mistakes. We have carefully corrected the linguistic errors in the revised *Manuscript*.

2. Regarding ROS-scavenging properties presented in figure 3, besides for panel a, it is not clear what are the used PtsaN-C concentration, and whether optimal enzyme-like activity is achieved in similar concentration for all.

Response: We apologize for the confusion in our original manuscript. Multienzyme-mimicking antioxidant properties of PtsaN-C, including CAT-like activity, POD-like activity, GPx-like activity and HO· scavenging property were measured under a concentration of 5 µg/ml. Accordingly, we have added corresponding details in the revised *Supplementary Information (Page 3, Line 2; Page 3, Line 18; Page 3, Line 24; Page 3, Line 30)*.

The rate of enzymatic reaction in the Michaelis–Menten mechanism is proportional to the concentration of the substrate molecules at low concentrations while it saturates at high concentrations [*J Chem Phys.* 2011 Apr 21;134(15):155101]. Turnover number (TON, mole of substrate converted per mole of active site), an indicator for catalytic efficiency, is widely used to assess the enzyme-like activity [*Adv Mater.* 2022 Nov;34(46):e2206208]. As the maximal reaction velocity (V_{max}) is only related to the properties of the enzyme and the enzymatic reaction conditions (such as temperature, pH , presence or absence of inhibitors, etc.) rather than concentration of the enzyme, TON is independent of enzyme concentration, which indicates enzyme activity does not correlate with enzyme concentration. Although the rate of reaction will be proportional to the concentration of the enzyme when the concentration of the substrate is sufficiently high, TON remain constant. Given that, we had not tested different concentrations of PtsaN-C, when we aimed to determine the multienzyme-mimicking antioxidant properties of PtsaN-C in this part. The TON of PtsaN-C was given in the *Supplementary Information Supplementary Table 6*. However, as enzyme activity might change due to the complexity of the biological systems involved, our in vitro and vivo carried out a wide range of concentration experiments to determine the optimal enzyme working concentration.

$$TON = V_{max}/[E]$$

[E], the molar concentration of the metal activation sites in nanozymes detected by ICP-MS;

V_{max} , maximal reaction velocity determined by Michaelis–Menten curves.

3. The authors state that the activity of PtsaN-C remains stable over 40°C. However, this temperature is irrelevant for the suggested application, and it should be noted that at normal body temperature the activity is about 70% while for catalase, naturally it is above 90%. Therefore, in that sense catalase (and the other enzymes) is superior since their activity is optimal at physiological temp.

Response: Thank you for the reviewer's comments. In the case of CAT-like activity, PtsaN-C was not as effective as natural catalase at 37°C. Natural catalase, exclusively responsible for the dismutation of H₂O₂ into H₂O and O₂, has desired catalytic efficiency and specificity under favourable conditions (amiable temperature, pH, solution etc.). However, the main disadvantage of natural catalase is that it is susceptible to the sophisticated biological environment especially disease states. The alteration of microenvironment after the ischemia/reperfusion injury remarkably restricts the catalytic activity of catalase. moreover, easy denaturation in the harsh environment, difficulties in long-term storage, and high cost in laborious preparation and purification also greatly restrict its applications. Our results suggested that the stability of PtsaN-C when exposed to *pH* and temperature variations was significantly greater than those of natural catalase (**Figure 3g-h**). Although catalytic efficiency of PtsaN-C was no better than catalase in a cell-free condition at 37°C, the stability of PtsaN-C confers advantage to contend with the complex biological environments upon injury. Our experiments also confirmed that PtsaN-C had overwhelming therapeutic effects over natural catalase in vivo (**Figure 6h, Supplementary Fig. 20**). Besides, the multienzyme-mimicking properties and good recyclability of PtsaN-C also made it superior to catalase. Therefore, PtsaN-C possesses distinct advantages during application in an organism.

4. Legend of main Figure 6 is cut off.

Response: Thank you very much for pointing this issue out. We apologize for this error and have adjusted the legend of Figure 6.

5. Error bars in figure 6l showing %EF and FS are surprisingly small. This is atypical, as usually there is higher variability between animals. Were these taken from different animals, or do they

represent technical replicates, of multiple measurements from the same animal? It is suggested to add individual data points.

Response: Sorry for the confusing. Echocardiography was taken from different animals. Data in the manuscript were presented with Mean \pm SEM. SEM (standard error of measurement) was calculated by the formula: $SEM = SD / \sqrt{n}$, with n being the size of the sample under examination. Hence, the greater n (sample size) was, the smaller SEM was. To allow for better comparison the difference of these two different manners, data with individual points were presented with Mean \pm SD and Mean \pm SEM respectively in **Figure R10**.

Figure R10. a EF (**left**) and FS (**right**) were presented with Mean \pm SD. **b** EF (**left**) and FS (**right**) were presented with Mean \pm SEM.

6. Please note that acronyms are not used without prior definition. Make sure to define all acronyms.

Response: Thank you very much for pointing this out. The full names of the acronyms have been provided in the revised *Manuscript*.

7. Figure 5a depicts the OGD/R procedure, depicting 6h “ischemia” and 24h reperfusion. However, in the methods section it is described as 8h with OGD and 12h reperfusion. Please make sure all information is consistent.

Response: Thank you very much for the kind reminding. We are sincerely sorry for this mistake. According to the published literatures [*BMC Med.* 2019;17:42; *Nat Commun.* 2022 Nov 9;13(1):6762], cells were first cultured for 8 hours with serum-free no glucose DMEM (Gibco) in an air-tight chamber with a humidified hypoxic atmosphere containing 5% CO₂ and 95% N₂ at 37 °C, and after exposure to oxygen glucose deprivation for 8 hours, the culture medium was replaced with serum and glucose-containing DMEM and transferred to a normal incubator for recovery for 12 hours. Accordingly, we have corrected the **Figure 5a** OGD/R procedure in the revised *Manuscript*.

Figure 5. Schematic diagram of experimental approach. H9C2 cells were cultured in serum-free no glucose DMEM in an air-tight chamber with a humidified hypoxic atmosphere containing 5% CO₂ and 95% N₂ at 37 °C for 8 hours. Then H9C2 cells were transferred to a normal incubator and cultured in serum and glucose-containing DMEM for recovery 12 hours.

8. There is room for improvement in the methods section (described in the SI). There are many missing details, and it should be more accurately described, and references should be included as suitable.

Response: Thank you for your kind suggestion. We have added more details in the methods section as marked with yellow background.

Overall, we would like to thank all the reviewers again for your time in reading and commenting on our manuscript. These detailed comments and suggestions indeed largely facilitate the revision of this paper. Thank you.

Reviewers' Comments:

Reviewer #1:

Remarks to the Author:

The authors have revised all parts. It is recommended that the manuscript be accepted

Reviewer #2:

Remarks to the Author:

The authors have revised the manuscript carefully according to the reviewers' comments. Some very important experimental and theoretical data were provided. After careful consideration, this manuscript is recommended to be accepted.

Reviewer #3:

Remarks to the Author:

In their revised manuscript the authors present a significantly improved manuscript.

The innovation is highlighted and better understood, the methodology section is more detailed and significant amount of data was added to support the narrative and to produce relevant information that strengthen the applicability of the developed particles as treatment for IRI.

This reviewer finds this work important and significant, with clear clinical, translational horizon.

The authors have rectified the reviewer's concerns and adequately replied to the comments.

Detailed Response to Reviewers

Reviewer 1

The authors have revised all parts. It is recommended that the manuscript be accepted.

Response: We thank the reviewer for the positive comments and supporting the publication of our manuscript.

Reviewer 2

The authors have revised the manuscript carefully according to the reviewers' comments. Some very important experimental and theoretical data were provided. After careful consideration, this manuscript is recommended to be accepted.

Response: Thanks very much for your positive comments and kind recommendation.

Reviewer 3

In their revised manuscript the authors present a significantly improved manuscript.

The innovation is highlighted and better understood, the methodology section is more detailed and significant amount of data was added to support the narrative and to produce relevant information that strengthen the applicability of the developed particles as treatment for IRI.

This reviewer finds this work important and significant, with clear clinical, translational horizon. The authors have rectified the reviewer's concerns and adequately replied to the comments.

Response: We deeply appreciate your careful review and kind consideration of our manuscript.

At last, we would like to sincerely thank all the reviewers again for your positive comments on our manuscript. Thank you.